# A Sober Look at Progress in Language Model Reasoning: Pitfalls and Paths to Reproducibility

**Andreas Hochlehnert**[1*]   **Hardik Bhatnagar**[1*]   **Vishaal Udandarao**[1,2∘]
**Samuel Albanie**   **Ameya Prabhu**[1†]   **Matthias Bethge**[1†]

[1]Tübingen AI Center, University of Tübingen    [2] University of Cambridge

🌐 Leaderboard        Code        Eval Logs

## Abstract

Reasoning has emerged as the next major frontier for language models (LMs), with rapid advances from both academic and industrial labs. However, this progress often outpaces methodological rigor, with many evaluations relying on benchmarking practices that lack transparency, robustness, or statistical grounding. In this work, we conduct a comprehensive empirical study and find that current mathematical reasoning benchmarks are highly sensitive to subtle implementation choices—including decoding parameters, random seeds, prompt formatting, and even hardware and software configurations. Performance gains reported in recent studies frequently hinge on unclear comparisons or unreported sources of variance. To address these issues, we propose a standardized evaluation framework with clearly defined best practices and reporting standards. Using this framework, we reassess recent methods and find that most reinforcement learning (RL) approaches yield only modest improvements—far below prior claims—and are prone to overfitting, especially on small-scale benchmarks like AIME'24. In contrast, supervised finetuning (SFT) methods show consistently stronger generalization in the settings we study. To foster reproducibility, we release all code, prompts, and model outputs, for reasoning benchmarks, establishing more rigorous foundations for future work.

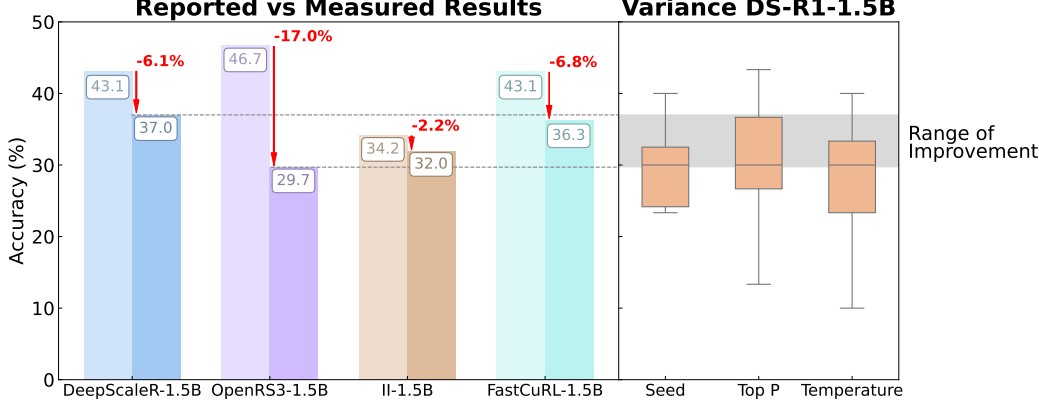

Figure 1: **The Sombre State of LM Reasoning for Math.** *(left)* when re-evaluating recent 1.5B reasoning-enhanced models on AIME-24 using a standardized framework (see Section 3), we find substantial drops to reported results in the original papers, *(right)* the observed improvements from recent methods (gray highlighted area) fall entirely within the variance range (orange box plots) of DeepSeek-R1 1.5B model performance. This suggests that these methods do not significantly outperform the base model—underscoring the importance of rigorous, multi-seed evaluation protocols for obtaining reliable performance estimates.

*equal contribution, ∘ core contributor, †equal advising

# 1 Introduction

> "The first principle is that you must not fool yourself, and you are the easiest person to fool."
>
> —Richard Feynman

*Reasoning* has become central to recent advances in large language models (LLMs), playing a key role in nearly all frontier systems (Jaech et al., 2024; Anthropic, 2025; OpenAI, 2025a; xAI, 2025; Meta-AI, 2025; DeepMind, 2025). Recent months have seen a surge of research focused on understanding and improving LLM reasoning, accompanied by several open-source tools and training strategies (see Li et al. (2025c) for a survey). This momentum has sparked optimism that building capable, competitive reasoning models may soon be within reach.

However, as evaluation practices shape the direction and perceived progress of the field (Liao et al., 2021), concerns around methodological rigor are growing. Non-reproducible or inconclusive evaluations can distort scientific understanding, misguide adoption, and skew future research priorities (Henderson et al., 2018; Marie et al., 2021; Musgrave et al., 2020; Prabhu et al., 2020; Andrychowicz et al., 2020; Colas et al., 2018). In the fast-moving area of LLM reasoning, where rapid publication cycles and benchmarking races are common, methodological shortcuts can quietly undermine progress. While concerns about reproducibility in LLM evaluations are well-documented (Reuel et al., 2024; Biderman et al., 2024), their persistence—especially in reasoning—calls for renewed scrutiny and higher standards.

Motivated by a growing number of inconsistent empirical claims across the reasoning landscape, we conduct a rigorous investigation into the current state of reasoning benchmarks, focusing specifically on mathematical reasoning—one of the most widely used testbeds for evaluating algorithmic advances in this space (HuggingFaceH4, 2024; AI-MO).

Our main finding is that many recent empirical conclusions may be overly optimistic and fail to generalize under careful re-evaluation. We identify a surprising degree of sensitivity in LLM-based reasoning pipelines to seemingly minor design choices—ranging from decoding parameters, prompt formatting, and random seeds to the hardware and software stacks used during evaluation (see Table 1). Particularly concerning is the instability introduced by small benchmark sizes: for example, AIME'24 and AMC'23 each contain only 30–40 examples, making performance metrics highly volatile—where even one question can shift Pass@1 by over 3 percentage points. This leads to substantial variance across seeds, often resulting in double-digit performance swings that challenge the reliability of published results. In Section 2, we systematically analyze the root causes of this instability, including sampling variance, decoding configurations, evaluation frameworks, and hardware heterogeneity. We show that these factors can significantly distort conclusions if not carefully controlled.

In Section 3, we propose a set of best practices aimed at improving reproducibility and rigor in reasoning benchmarks. We also re-evaluate recent techniques using a standardized and reproducible evaluation stack. Our findings are sobering—reinforcement learning (RL) applied to distillation-based models such as DeepSeek-R1 yields little to no statistically significant gains. Some methods, such as OpenRS, show promising results in original reports, but fail to hold up under repeated evaluation. RL training on base models like Qwen2.5 Math does show stronger performance, but still often underperforms instruction-tuned counterparts.[1] Furthermore, RL-trained models exhibit significant performance drops on newer benchmarks such as AIME'25, echoing patterns of test set overfitting or "hill-climbing" observed in prior work (Golchin & Surdeanu, 2023; Roberts et al., 2023; Dominguez-Olmedo et al., 2024). In contrast, supervised fine-tuning (SFT) continues to deliver stable, generalizable improvements across benchmarks, underscoring its maturity as a training paradigm. These observations point to a critical need for more reliable and standardized evaluation protocols.

Taken together, in this work, we aim to provide not only a clearer assessment of where current methods stand, but also the tools and practices needed to make reasoning evaluation more transparent, robust, and reproducible. To this end, we open-source all code, prompts, and outputs to facilitate fair and accountable progress in this increasingly important area.

---

[1]We note that OpenReasoner-Zero is a consistent exception, achieving competitive performance.

## 2 Exploring the Design Space of Reasoning: What Matters Most?

Recent reasoning-focused language models are evaluated under highly heterogeneous conditions—including differences in evaluation frameworks and hardware, number of random seeds, temperature, and nucleus sampling parameters (top_p) (see Table 1). While prior work has examined the effect of sampling parameters in multiple-choice (Renze, 2024) and coding tasks (Arora et al., 2024), the influence of these choices remains underexplored for open-ended reasoning models—particularly those trained with reinforcement learning. In this section, we systematically assess how these evaluation design choices affect reported performance, and highlight the sources of variance that most impact the reliability of results.

### 2.1 Experimental Setup

We adopt a consistent experimental setup throughout this section, unless otherwise stated. Our analysis includes nine widely used models grouped into two commonly benchmarked size classes: 1.5B and 7B parameters. For the 1.5B class, we evaluate: DeepSeek-R1-Distill-1.5B (DeepSeek-AI, 2025), DeepScaleR-1.5B (Luo et al., 2025), II-1.5B-Preview (Intelligent Internet, 2025) , OpenRS1-1.5B, OpenRS2-1.5B, and OpenRS3-1.5B (Dang & Ngo, 2025). Note that DeepScaleR-1.5B, II-1.5B-Preview, and the OpenRS models are all initialized from DeepSeek-R1-Distill-1.5B and subsequently finetuned via reinforcement learning (e.g., GRPO (Shao et al., 2024)) to enhance mathematical reasoning capabilities. For the 7B class, we evaluate: DeepSeek-R1-Distill-7B, S1.1-7B (Muennighoff et al., 2025), and OpenThinker-7B (Open Thoughts, 2025). Both S1.1-7B and OpenThinker-7B are finetuned Qwen2.5-7B-Instruct models (Yang et al., 2024a), trained using supervised learning on reasoning traces derived from DeepSeek-R1. All models are benchmarked on three widely used datasets: AIME'24 (AI-MO), AMC'23 (AI-MO, 2024), and MATH500 (Hendrycks et al., 2021), using the Pass@1 metric. Each result is averaged over multiple seeds and obtained on a standardized software stack (throguh a Docker image), and hardware with the following configuration: one 40 GB A100 GPU, an AMD 7302 32-core CPU, and 1TB RAM. All experiments were run using lighteval (Fourrier et al., 2023) with the vllm backend (Kwon et al., 2023).

**Sampling Parameters:** To systematically compare the impact of sampling parameters on accuracy, all experiments in this section were performed with a standardized configuration: temperature=0.8, top_p=0.9, and both max_model_len and max_new_tokens set to 32,768 tokens. This context length matches the limits of models such as OpenThinker-7B and S1.1-7B, although certain models (e.g., DeepSeek) support longer sequences of up to 131,072 tokens. We chose this standardized evaluation length to ensure comparability, with a detailed analysis of the influence of completion length presented in Figure 6. Unless otherwise specified, results in this section are averaged over 10 random seeds for AIME'24 and AMC'23, and 3 seeds for MATH500, following the recommendations from Section 2.2.1.

### 2.2 Seed Variance in Evaluation

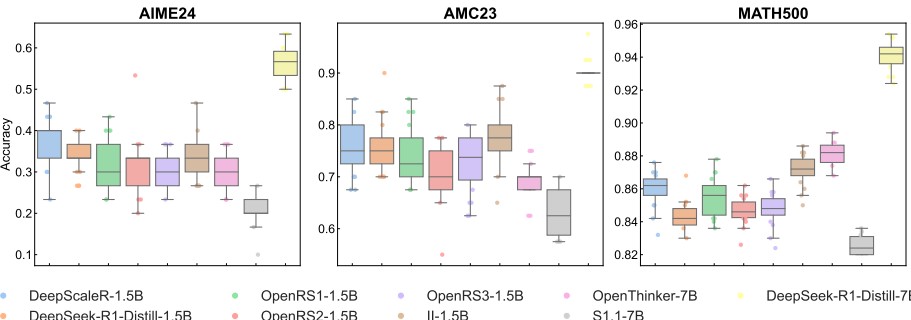

Figure 2: **Accuracy varies significantly across random seeds.** We find significantly high Pass@1 variation across 20 different random seeds for nine models on AIME'24, AMC'23, and MATH500. Variance is particularly high on AIME'24 (upto 15%) and AMC'23 (upto 13%) due to the small number of test samples, highlighting instability of single-seed evaluations.

Table 1: **Taxonomy of current open-weight reasoning models.** For each model, we report the *base* model it was post-trained from and the exact type of post-training *algorithm* applied (RL vs SFT). Further, we note the *evaluation framework* that the original paper uses for reporting results along with the exact *temperature*, *generation sequence length*, and *top_p* sampling parameters used for AIME-24 evaluation, with the number of generations used for computing Pass@1 (*K*). It is evident that there is no clear standardization across different models with respect to evaluation frameworks used and the sampling parameters. This motivates the need to closely scrutinize the evaluations of current reasoning models.

| Model | Algorithm | Base | Framework | Temp | Top_p | Seq. Len | K |
|---|---|---|---|---|---|---|---|
| DeepSeek-R1-Distill-1.5B | SFT | Qwen2.5-Math-1.5B | – | 0.6 | 0.95 | 32,768 | 64 |
| DeepSeek-R1-Distill-7B | SFT | Qwen2.5-Math-7B | – | 0.6 | 0.95 | 32,768 | 64 |
| DeepSeek-R1-Distill-14B | SFT | Qwen2.5-14B | – | 0.6 | 0.95 | 32,768 | 64 |
| DeepSeek-R1-Distill-32B | SFT | Qwen2.5-32B | – | 0.6 | 0.95 | 32,768 | 64 |
| OpenThinker-32B | SFT | Qwen2.5-32B-Instruct | evalchemy | 0.7 | 0.8 | 32,768 | 5 |
| Bespoke-Stratos-32B | SFT | Qwen2.5-32B-Instruct | evalchemy | 0.7 | 0.8 | 32,768 | 5 |
| Bespoke-Stratos-7B | SFT | Qwen2.5-7B-Instruct | evalchemy | 0.7 | 0.8 | 32,768 | 5 |
| s1.1-7B | SFT | Qwen2.5-7B-Instruct | lm-eval-harness | 0 | – | 32,768 | 64 |
| s1.1-32B | SFT | Qwen2.5-32B-Instruct | lm-eval-harness | 0 | – | 32,768 | 64 |
| LIMO | SFT | Qwen2.5-32B-Instruct | math-eval-harness | 0 | 1 | 32,768 | 1 |
| MiniMath-R1-1.5B | SFT | DeepSeek-R1-Distill-1.5B | oumi-ai | – | – | – | – |
| Light-R1-7B | SFT | DeepSeek-R1-Distill-7B | verl | 0.6 | 0.95 | 32,768 | – |
| DeepScaleR-1.5B-Preview | RL | DeepSeek-R1-Distill-1.5B | verl | 0.6 | 0.95 | 32,768 | 16 |
| L1-Exact | RL | DeepSeek-R1-Distill-1.5B | verl | 0.6 | 0.95 | 8,000 | – |
| L1-Max | RL | DeepSeek-R1-Distill-1.5B | verl | 0.6 | 0.95 | 8,000 | – |
| Open-RS1 | RL | DeepSeek-R1-Distill-1.5B | lighteval | 0.6 | 0.95 | 32,768 | 32 |
| Open-RS2 | RL | DeepSeek-R1-Distill-1.5B | lighteval | 0.6 | 0.95 | 32,768 | 32 |
| Open-RS3 | RL | DeepSeek-R1-Distill-1.5B | lighteval | 0.6 | 0.95 | 32,768 | 32 |
| II-Thought-1.5B-Preview | RL | DeepSeek-R1-Distill-1.5B | evalscope | 0.6 | 0.95 | 32,768 | 64 |
| Oat-Zero-1.5B | RL | Qwen2.5-Math-1.5B | custom | 0 | 1 | 3,000 | 1 |
| Oat-Zero-7B | RL | Qwen2.5-Math-7B | custom | 0 | 1 | 3,000 | 1 |
| STILL-3-1.5B-preview | RL | DeepSeek-R1-Distill-1.5B | custom | 0.6 | 0.95 | 32,768 | 5 |
| FastCurl-1.5B-Preview | RL | DeepSeek-R1-Distill-1.5B | verl | 0.6 | 0.95 | 32,768 | 16 |
| LIMR | RL | Qwen2.5-Math-7B | custom | 0.4 | 0.95 | 3,072 | 4 |
| SimpleRL-Zoo-7B | RL | Qwen2.5-7B | verl | 1 | 0.95 | 16,000 | 32 |
| SimpleRL-Zoo-Math-7B | RL | Qwen2.5-Math-7B | verl | 1 | 0.95 | 16,000 | 32 |
| OpenReasoner-Zero-7B | RL | Qwen2.7-7B | – | 1 | 1 | – | 16 |
| Eurus2 Prime | RL | Qwen2.5-Math-7B | custom | 0 | 1 | 4,096 | – |

We begin by analyzing the variance induced purely by the random seed used during evaluation—an aspect often neglected in benchmarking practices. While recent work calls for statistical rigor (e.g., using error bars and multiple runs) (Bowyer et al., 2025; Biderman et al., 2024; Madaan et al.), evaluations frequently rely on single-seed runs, obscuring potential variability. We assess the seed-induced variance across 20 independent evaluation runs for each of the nine models. Results are shown in Figure 2.

**Key Insight.** Pass@1 values show surprisingly high standard deviation—ranging from 5 to 15 percentage points across seeds. This issue is particularly severe for AIME'24 and AMC'23, which have only 30 and 40 test samples respectively. A change in just one question shifts Pass@1 by 2.5–3.3 percentage points.

> **Takeaway 1** Single-seed evaluations on small datasets are highly unstable. Accurate reporting requires averaging over multiple seeds.

> **Takeaway 2** Small datasets such as AIME24 (30 samples) make model comparisons unreliable, as solving just one extra question already shifts pass@1 by 3%. Variance from sampling parameters or random seeds can easily cause fluctuations of 1–2 correct answers, leading to unstable rankings – especially when models cluster around 30% performance.

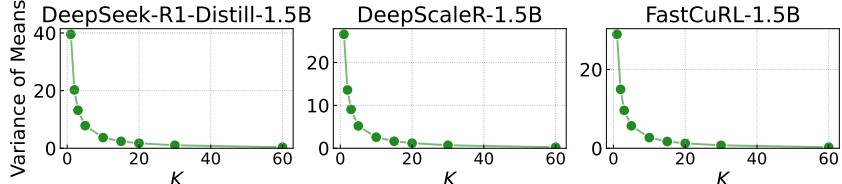

Figure 3: **Bootstrapped seed averaging is reliable only beyond a threshold.** We plot the variance of Mean Pass@1 scores on AIME'24 when averaging over $K = 1$ to $K = 60$ seed runs, finding that the variance is extremely high for small $K$ and significantly reduced by $K = 30$. This suggests that using multi-seed evaluations ($K \geq 30$) would yield more stable estimates. For results on AMC23 and MATH500 see Figures 12 and 14 respectively.

### 2.2.1 Can Bootstrapping Improve Mean Estimates?

To mitigate high variance, recent work has adopted bootstrapping—averaging multiple evaluation runs to stabilize results. For example, DeepSeek reports Pass@1 over 64 runs, while DeepScaleR uses 16. We study the effectiveness of this approach by bootstrapping estimates for AIME'24 using 1 to 60 evaluation runs. Figure 3 shows that while variance is extreme for $K = 1$ and still large for $K = 5$, it reduces sharply for $K \geq 30$. Further analysis of variance across additional datasets is presented Figures 12 and 14.

> **Takeaway 3** Bootstrapping over 30 runs substantially stabilizes Pass@1 estimates and should be considered a minimal standard for reliable evaluation.

### 2.2.2 Variance from Sampling Parameters: Temperature and top-p

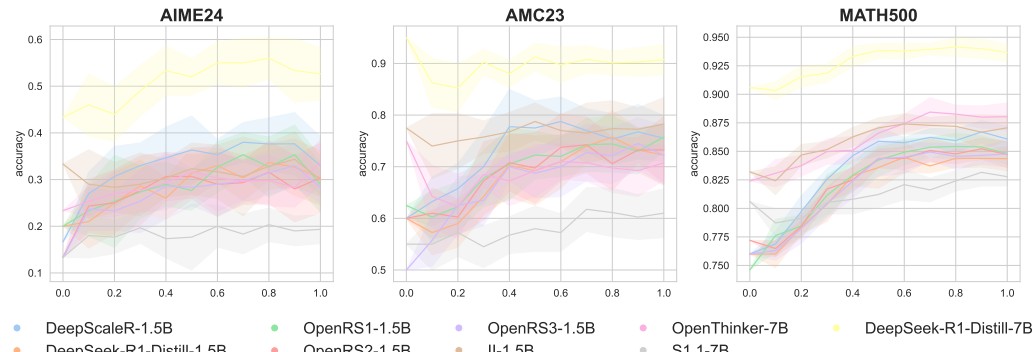

Figure 4: **Higher temperatures yield better accuracies.** We find across all three datasets, higher temperatures produce better peak accuracy but introduce instability, revealing a tradeoff between performance and reproducibility. Results obtained by varying temperature from 0 to 1 in increments of 0.1, while keeping `top_p` fixed at 0.9.

Reducing the temperature or increasing the nucleus sampling parameter (`top_p`) improves the accuracy of performance estimates without incurring additional computational cost. Figure 4 illustrate the impact of temperature and Figure 5 show that of `top_p` across multiple models and datasets. Notably, a more reproducible estimate is associated with significant drops in measured performance, highlighting a consistent tradeoff between reproducibility and high performance. We recommend optimizing the temperature for performance, and comparing the best parameter per model.

> **Takeaway 4** Temperature and `top_p` can introduce substantial performance variation—especially on small benchmarks—and should be set to each model's optimal values to ensure fair and stable evaluation.

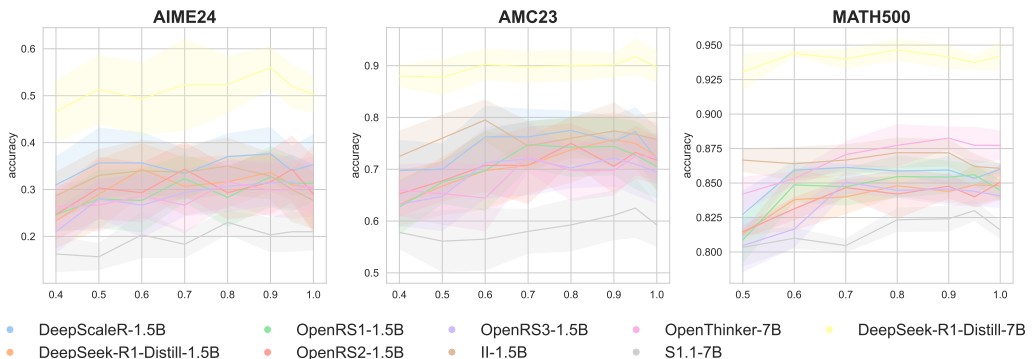

Figure 5: **Higher `top_p` values improve performance at no cost to stability.** Across all datasets, we find that higher `top_p` values generally improve performance while preserving similar amounts of variance as lower `top_p` values. Results were obtained by varying `top_p` from 0 to 1 in increments of 0.1, while holding the temperature constant at 0.8.

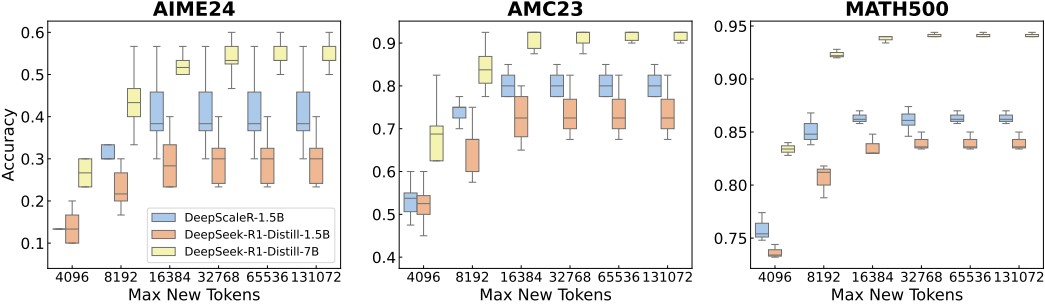

Figure 6: **Models are extremely sensitive to output token lengths.** We sweep across different `max_new_tokens` (number of tokens that models are allowed to generate) for DeepScaleR-1.5B and DeepSeek-R1-Distill-1.5B/7B on three datasets and find that they are heavily sensitive to output length limits, with premature truncation degrading the performance.

## 2.3 Effect of Prompt Format and Context Length

**Maximum Output Tokens.** Figure 6 shows that reducing `max_new_tokens` harms performance—especially on long-form problems. This sensitivity varies by model and dataset. Although reducing this setting lowers cost, it may induce premature stopping, leading to incorrect answers.

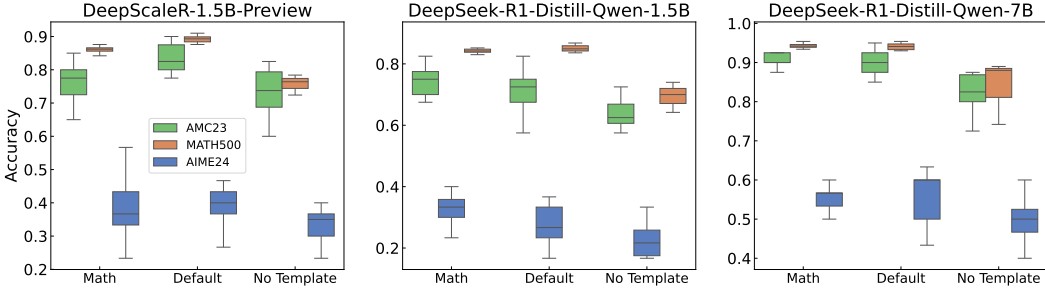

Figure 7: **Using no prompt templates yields worse performance.** We compare Pass@1 scores across three prompt formats: (1) math-specific prompt with chat template, (2) default chat template only, and (3) no template. Instruction-tuned models perform best with structured prompts and templates; omitting templates leads to consistent performance drops.

**Prompt Format.** Prompt formatting has a measurable impact on accuracy. As shown in Figure 7, models perform best when using math-specific prompts and their native chat templates. Omitting templates leads to performance drops, particularly for instruction-tuned models. We compare accuracy under three different prompt settings (see Table 5): (1)

a math-specific prompt formatted using the model's chat template, (2) only the model's chat template with no additional prompt, and (3) no template at all, i.e., the question without any special tokens or instructions. Interestingly, while base models like Qwen2.5-Math may benefit from prompt-free setups (Liu et al., 2025b), instruction-tuned models rely heavily on format alignment. Thus, maintaining consistent and format-aware prompting is essential for maximizing instruction-tuned model performance.

> **Takeaway 5** It is critical to use large generation context lengths to avoid output truncation which can degrade performance; further, using correct prompt formats and chat templates is important for extracting best model performance.

## 3 Way Forward: Standardization in Evaluations

In this section, we standardize evaluation frameworks, propose best practices, and comprehensively evaluate existing methods.

### 3.1 Standardization Procedure

We adopt a largely consistent experimental setup with prior work, with the key difference being our use of publicly accessible cloud instances from Runpod[2]. Each instance is equipped with a single A100 PCIe GPU, 8 vCPUs, and 128 GB of RAM. We evaluate all models listed in Table 2 across six benchmarks: AIME'24 (AI-MO), AIME'25 (Lin, 2025), AMC'23 (Knovel Engineering, 2025), MATH500 (HuggingFaceH4, 2024), Minerva (Lewkowycz et al., 2022), and OlympiadBench (He et al., 2024). All experiments are conducted using the LightEval framework (Fourrier et al., 2023) (0.8.1) with a vLLM backend, repeated across ten random seeds for AIME'24, AIME'25, AMC'23 and three random seeds for the rest. Depending on the base model architecture, we set the maximum number of new tokens (e.g., 4096 for QwenMath-based models), apply optimal hyperparameters, and use the standardized chat template except for base models. LightEval's LaTeX-based answer extraction and evaluation pipeline ensures reliable and consistent result parsing and correctness matching, similar to `math-verify`.

**Statistical Significance Testing.** To rigorously assess whether RL-trained models genuinely outperform their baselines, we employ paired sample-wise statistical tests rather than simply comparing mean accuracies. For each model-baseline pair, we: (1) compute the average accuracy per problem instance across all random seeds, (2) align these sample-wise accuracies between the RL/SFT model and the baseline from which it was trained, and (3) conduct one-sided paired tests to evaluate whether the RL model achieves systematically higher performance. We employ both parametric (paired t-test) and non-parametric (Wilcoxon signed-rank) tests to ensure robustness to distributional assumptions. The null hypothesis is that there is no improvement from RL/SFT training, with the alternative hypothesis that the RL/SFT model performs better than its baseline. We report results as statistically significant at two thresholds: $p < 0.01$ (marked with [*]) and $p < 0.001$ (marked with [**]). This conservative approach ensures that reported improvements represent genuine performance gains rather than statistical artifacts from multiple testing or random variation.

### 3.2 A Sober Look: Results

We present experimental results in Table 2, and analyze different aspects of the results.

**RL-training on R1-Distill** We evaluated several reinforcement learning (RL) approaches (e.g., GRPO) using the DeepSeek R1-Distill-1.5B model. We first observe that none of the L1 models (Aggarwal & Welleck, 2025) outperformed the original DeepSeek R1-Distill baseline — an expected outcome given that L1 training prioritized smaller output length over accuracy. OpenRS (Dang & Ngo, 2025) reported strong gains (10–15%) on AIME, AMC, and OlympiadBench. However, our replication showed no statistically significant

---

[2] https://www.runpod.io/pricing

| Model | AIME'24 | AIME'25 | AMC'23 | MATH500 | Minerva | Olympiad |
|---|---|---|---|---|---|---|
| Based on: Deepseek R1 Distill Qwen 1.5B (RL) | | | | | | |
| R1-Distill (DeepSeek-AI, 2025) | $28.7_{\pm 4.8}$ | $22.3_{\pm 5.2}$ | $71.5_{\pm 3.9}$ | $84.9_{\pm 0.3}$ | $30.5_{\pm 1.0}$ | $52.4_{\pm 0.4}$ |
| L1-Exact (Aggarwal & Welleck, 2025) | $24.4_{\pm 3.3}$ | $22.3_{\pm 4.2}$ | $70.5_{\pm 3.7}$ | $86.6_{\pm 0.8}$ | $31.5_{\pm 1.7}$ | $52.5_{\pm 1.3}$ |
| L1-Max (Aggarwal & Welleck, 2025) | $27.7_{\pm 4.2}$ | $21.0_{\pm 5.0}$ | $73.2_{\pm 6.0}$ | $84.7_{\pm 0.1}$ | $33.3_{\pm 0.9}$ | $52.3_{\pm 0.6}$ |
| Open-RS1 (Dang & Ngo, 2025) | $28.9_{\pm 6.0}$ | $21.3_{\pm 4.2}$ | $75.0_{\pm 3.3}$ | $85.1_{\pm 0.8}$ | $30.4_{\pm 0.2}$ | $53.2_{\pm 1.9}$ |
| Open-RS2 (Dang & Ngo, 2025) | $31.3_{\pm 7.7}$ | $22.7_{\pm 5.6}$ | $73.0_{\pm 5.7}$ | $84.1_{\pm 0.2}$ | $29.2_{\pm 1.1}$ | $53.7_{\pm 0.6}$ |
| Open-RS3 (Dang & Ngo, 2025) | $29.7_{\pm 4.6}$ | $24.7_{\pm 6.5}$ | $69.2_{\pm 5.5}$ | $84.2_{\pm 1.1}$ | $28.6_{\pm 2.3}$ | $51.8_{\pm 0.8}$ |
| STILL-3 (Min et al., 2024) | $34.7_{\pm 5.5}$ | $24.0_{\pm 6.4}$ | $72.5_{\pm 5.4}$ | $86.6_{\pm 1.9}$ | $30.0_{\pm 0.6}$ | $53.9_{\pm 1.5}$ |
| ZR1-1.5B (Zyphra, 2025) | $30.3_{\pm 4.6}$ | $24.0_{\pm 3.3}$ | $78.8_{\pm 4.0}$ | $86.6_{\pm 0.9}$ | $32.1_{\pm 0.9}$ | $56.4_{\pm 0.4}^{**}$ |
| II-Thought (Intelligent Internet, 2025) | $32.0_{\pm 5.9}$ | $24.0_{\pm 4.1}$ | $79.5_{\pm 5.1}^{*}$ | $86.6_{\pm 0.6}$ | $31.7_{\pm 0.6}$ | $54.9_{\pm 0.4}^{*}$ |
| DeepScaleR (Luo et al., 2025) | $37.0_{\pm 6.6}$ | $30.3_{\pm 4.3}^{**}$ | $76.2_{\pm 4.6}$ | $87.8_{\pm 1.0}^{**}$ | $31.0_{\pm 1.5}$ | $55.5_{\pm 1.1}^{**}$ |
| FastCuRL (Song et al., 2025) | $36.3_{\pm 4.6}^{*}$ | $26.5_{\pm 3.7}$ | $76.9_{\pm 4.8}$ | $87.5_{\pm 1.0}^{**}$ | $30.8_{\pm 1.6}$ | $56.7_{\pm 0.8}^{**}$ |
| DisCO-logL (Li et al., 2025a) | $39.0_{\pm 4.5}$ | $31.7_{\pm 4.8}^{*}$ | $78.8_{\pm 3.4}^{*}$ | $87.4_{\pm 1.1}^{*}$ | $32.7_{\pm 1.3}$ | $58.6_{\pm 0.3}^{**}$ |
| Nemotron-RR (Mingjie Liu, 2025) | $48.3_{\pm 6.3}^{**}$ | $33.0_{\pm 5.1}^{*}$ | $87.0_{\pm 4.8}^{**}$ | $91.0_{\pm 0.5}^{**}$ | $37.5_{\pm 1.6}^{**}$ | $62.8_{\pm 0.9}^{**}$ |
| Based on: Deepseek R1 Distill Qwen 7B (RL) | | | | | | |
| R1-Distill (DeepSeek-AI, 2025) | $52.3_{\pm 6.3}$ | $39.0_{\pm 5.9}$ | $91.5_{\pm 2.7}$ | $94.1_{\pm 0.3}$ | $40.1_{\pm 0.4}$ | $67.3_{\pm 0.1}$ |
| Sky-T1 (mini) (NovaSky, 2025) | $51.3_{\pm 4.5}$ | $39.0_{\pm 5.7}$ | $90.0_{\pm 3.3}$ | $93.3_{\pm 1.0}$ | $41.8_{\pm 1.2}$ | $67.7_{\pm 0.9}$ |
| DisCO-logL (Li et al., 2025a) | $51.3_{\pm 3.9}$ | $41.3_{\pm 4.5}$ | $92.2_{\pm 2.2}$ | $94.0_{\pm 1.3}$ | $44.7_{\pm 1.1}^{**}$ | $67.6_{\pm 0.3}$ |
| Skywork-OR1 (He et al., 2025) | $60.7_{\pm 4.7}^{*}$ | $49.7_{\pm 6.2}^{*}$ | $94.0_{\pm 3.8}$ | $95.7_{\pm 0.9}^{*}$ | $43.0_{\pm 1.0}^{*}$ | $73.6_{\pm 0.9}^{**}$ |
| Based on: Deepseek R1 Distill Qwen 7B (SFT) | | | | | | |
| R1-Distill (DeepSeek-AI, 2025) | $52.3_{\pm 6.3}$ | $39.0_{\pm 5.9}$ | $91.5_{\pm 2.7}$ | $94.1_{\pm 0.3}$ | $40.1_{\pm 0.4}$ | $67.3_{\pm 0.1}$ |
| Light-R1 (Wen et al., 2025b) | $53.0_{\pm 4.8}$ | $41.0_{\pm 3.5}$ | $90.0_{\pm 3.1}$ | $93.5_{\pm 0.5}$ | $41.3_{\pm 1.3}$ | $68.0_{\pm 1.2}$ |
| Based on: Qwen2.5 Math 1.5B (RL) | | | | | | |
| Math (Base) (Yang et al., 2024b) | $11.3_{\pm 3.6}$ | $5.7_{\pm 2.7}$ | $44.0_{\pm 4.9}$ | $51.7_{\pm 5.5}$ | $11.3_{\pm 2.2}$ | $26.0_{\pm 0.6}$ |
| Oat-Zero (Liu et al., 2025a) | $16.0_{\pm 3.2}$ | $6.7_{\pm 3.4}$ | $52.5_{\pm 2.9}$ | $73.5_{\pm 1.7}^{**}$ | $26.3_{\pm 0.8}^{**}$ | $37.2_{\pm 1.3}^{**}$ |
| Math (Instruct) (Yang et al., 2024b) | $12.0_{\pm 1.7}$ | $11.7_{\pm 5.7}^{*}$ | $54.8_{\pm 5.3}$ | $74.7_{\pm 0.5}^{**}$ | $26.7_{\pm 1.8}^{**}$ | $37.9_{\pm 0.2}^{**}$ |
| LUFFY-Zero (Yan et al., 2025a) | $15.9_{\pm 4.9}$ | $11.7_{\pm 2.8}$ | $54.5_{\pm 5.4}$ | $77.6_{\pm 1.2}^{**}$ | $27.2_{\pm 0.4}^{**}$ | $42.3_{\pm 0.5}^{**}$ |
| Based on: Qwen2.5 Math 7B (RL) | | | | | | |
| Math (Base) (Yang et al., 2024b) | $20.7_{\pm 3.8}$ | $8.7_{\pm 3.9}$ | $56.2_{\pm 5.7}$ | $64.3_{\pm 0.5}$ | $17.3_{\pm 1.9}$ | $29.0_{\pm 0.5}$ |
| SimpleRL-Zoo (Zeng et al., 2025b) | $22.7_{\pm 5.2}$ | $10.7_{\pm 3.4}$ | $62.2_{\pm 3.6}$ | $76.9_{\pm 1.8}^{**}$ | $30.1_{\pm 2.8}^{**}$ | $39.3_{\pm 0.6}^{**}$ |
| LIMR (Li et al., 2025b) | $30.7_{\pm 3.2}$ | $7.8_{\pm 3.3}$ | $62.2_{\pm 3.4}$ | $76.5_{\pm 0.4}^{**}$ | $34.9_{\pm 1.3}^{**}$ | $39.3_{\pm 0.9}^{**}$ |
| Oat-Zero (Liu et al., 2025a) | $28.0_{\pm 3.1}$ | $8.8_{\pm 2.5}$ | $66.2_{\pm 3.6}$ | $79.4_{\pm 0.3}^{**}$ | $34.4_{\pm 1.4}^{**}$ | $43.8_{\pm 1.1}^{**}$ |
| Sky-T1 (NovaSky, 2025) | $19.3_{\pm 2.6}$ | $21.0_{\pm 3.9}^{*}$ | $67.2_{\pm 3.6}^{**}$ | $85.2_{\pm 0.5}^{**}$ | $34.7_{\pm 1.1}^{**}$ | $52.1_{\pm 0.8}^{**}$ |
| Math (Instruct) (Yang et al., 2024b) | $15.7_{\pm 3.9}$ | $10.7_{\pm 3.8}$ | $67.0_{\pm 3.9}$ | $82.9_{\pm 0.1}^{**}$ | $35.0_{\pm 0.6}^{**}$ | $41.3_{\pm 0.9}^{**}$ |
| LUFFY-Zero (Yan et al., 2025a) | $26.7_{\pm 6.5}$ | $23.3_{\pm 5.7}^{*}$ | $73.5_{\pm 3.9}^{**}$ | $87.9_{\pm 0.1}^{**}$ | $34.1_{\pm 2.2}^{**}$ | $54.9_{\pm 0.3}^{**}$ |
| Based on: Qwen2.5 1.5B (RL) | | | | | | |
| Qwen (Base) (Yang et al., 2024a) | $0.0_{\pm 0.0}$ | $0.0_{\pm 0.0}$ | $2.5_{\pm 2.5}$ | $3.3_{\pm 1.5}$ | $1.8_{\pm 0.4}$ | $1.5_{\pm 0.5}$ |
| SimpleRL-Zoo (Zeng et al., 2025b) | $0.3_{\pm 1.1}$ | $0.3_{\pm 1.1}$ | $13.2_{\pm 4.7}^{*}$ | $12.0_{\pm 6.5}^{**}$ | $4.0_{\pm 2.4}^{*}$ | $4.2_{\pm 2.0}^{**}$ |
| Qwen (Instruct) (Yang et al., 2024a) | $1.3_{\pm 1.7}$ | $0.7_{\pm 1.4}$ | $26.2_{\pm 4.8}^{**}$ | $58.1_{\pm 1.4}^{**}$ | $19.4_{\pm 1.3}^{**}$ | $20.4_{\pm 0.5}^{**}$ |
| Based on: Qwen2.5 7B (RL) | | | | | | |
| Qwen (Base) (Yang et al., 2024a) | $8.0_{\pm 3.2}$ | $3.7_{\pm 3.7}$ | $35.8_{\pm 4.9}$ | $59.7_{\pm 0.3}$ | $21.4_{\pm 1.9}$ | $27.0_{\pm 1.0}$ |
| SimpleRL-Zoo (Zeng et al., 2025b) | $14.0_{\pm 2.1}$ | $4.3_{\pm 2.7}$ | $58.0_{\pm 1.6}^{**}$ | $77.9_{\pm 0.8}^{**}$ | $33.0_{\pm 0.2}^{**}$ | $39.0_{\pm 0.1}^{**}$ |
| Open Reasoner Zero (Hu et al., 2025) | $19.7_{\pm 2.9}$ | $15.7_{\pm 2.7}$ | $59.5_{\pm 4.5}^{**}$ | $83.9_{\pm 1.1}^{**}$ | $31.6_{\pm 1.3}^{**}$ | $47.6_{\pm 1.7}^{**}$ |
| Qwen (Instruct) (Yang et al., 2024a) | $12.3_{\pm 3.2}$ | $7.3_{\pm 3.4}$ | $52.8_{\pm 4.8}^{**}$ | $77.3_{\pm 0.8}^{**}$ | $34.9_{\pm 1.0}^{**}$ | $38.9_{\pm 0.9}^{**}$ |
| Based on: Qwen2.5 7B Instruct (SFT) | | | | | | |
| Qwen (Instruct) (Yang et al., 2024a) | $12.3_{\pm 3.2}$ | $7.3_{\pm 3.4}$ | $52.8_{\pm 4.8}$ | $77.3_{\pm 0.8}$ | $34.9_{\pm 1.0}$ | $38.9_{\pm 0.9}$ |
| Eurus2 Prime (Cui et al., 2025) | $18.7_{\pm 1.7}$ | $14.3_{\pm 1.6}$ | $64.8_{\pm 4.0}$ | $80.1_{\pm 0.1}^{**}$ | $37.5_{\pm 1.0}^{**}$ | $43.9_{\pm 0.3}^{**}$ |
| s1.1 (Muennighoff et al., 2025) | $19.0_{\pm 3.2}$ | $21.0_{\pm 5.5}^{*}$ | $59.5_{\pm 3.7}$ | $80.8_{\pm 0.6}^{*}$ | $37.5_{\pm 1.1}$ | $48.2_{\pm 1.4}^{**}$ |
| Bespoke Stratos (Bespoke Labs, 2024) | $20.3_{\pm 4.3}$ | $18.0_{\pm 4.8}$ | $60.2_{\pm 4.9}$ | $84.7_{\pm 0.5}^{**}$ | $39.1_{\pm 1.3}^{*}$ | $51.9_{\pm 1.1}^{**}$ |
| OpenThinker (Open Thoughts, 2025) | $30.5_{\pm 6.2}^{**}$ | $26.0_{\pm 4.4}^{**}$ | $71.4_{\pm 3.9}^{**}$ | $88.3_{\pm 1.4}^{**}$ | $37.9_{\pm 3.8}^{**}$ | $55.6_{\pm 1.4}^{**}$ |
| OpenR1 (Hugging Face, 2025) | $48.3_{\pm 8.9}^{**}$ | $35.5_{\pm 4.2}^{**}$ | $86.0_{\pm 4.5}^{**}$ | $93.5_{\pm 0.7}^{**}$ | $41.2_{\pm 1.3}^{**}$ | $67.4_{\pm 1.3}^{**}$ |
| OpenThinker2 (Open Thoughts, 2025) | $53.0_{\pm 4.6}^{**}$ | $41.0_{\pm 5.0}^{**}$ | $87.0_{\pm 3.5}^{**}$ | $94.4_{\pm 0.7}^{**}$ | $42.0_{\pm 1.5}^{**}$ | $70.6_{\pm 1.0}^{**}$ |

Table 2: **A Standardized and Sober Compilation of LM-Reasoning Results.** We report Pass@1 accuracy (mean $\pm$ std) of all models across six math reasoning benchmarks under a standardized evaluation setup. RL- and SFT-based variants are evaluated relative to their respective base or instruction-tuned models. Main takeaways are that (1) RL-trained methods do not yield meaningful performance gains, (2) SFT on reasoning traces yields significant generalization. Note that $^*$ statistically significant at $p < 0.01$; $^{**}$ statistically significant at $p < 0.001$ (paired t-test relative to base model).

improvements over the R1 - Distill baseline. Same case held for Still-3 and Light-R1 model, which showed no significant improvement over the R1-Distill baseline. II-Thought yields modest improvements across benchmarks, especially over AIME'24 but the observed gains

| Model | AIME'24 | AIME'25 | AMC'23 | MATH500 | Minerva | Olympiad |
|---|---|---|---|---|---|---|
| Based on: Qwen2.5 32B (RL) | | | | | | |
| Qwen (Base) (Yang et al., 2024a) | $8.0_{\pm5.5}$ | $2.3_{\pm2.2}$ | $36.5_{\pm6.8}$ | $62.8_{\pm3.5}$ | $28.9_{\pm2.2}$ | $29.5_{\pm2.0}$ |
| DAPO (Yu et al., 2025) | $43.0_{\pm4.0}$** | $32.3_{\pm6.1}$** | $88.5_{\pm3.6}$** | $91.0_{\pm0.3}$** | $40.0_{\pm0.6}$** | $60.9_{\pm1.1}$** |
| Qwen (Instruct) (Yang et al., 2024a) | $15.7_{\pm4.7}$ | $13.0_{\pm4.6}$ | $60.8_{\pm5.7}$ | $81.6_{\pm0.5}$ | $40.3_{\pm1.3}$ | $46.7_{\pm1.3}$ |
| Based on: Qwen QwQ 32B (RL) | | | | | | |
| QwQ-32B (Team, 2025) | $76.3_{\pm3.3}$ | $69.0_{\pm4.5}$ | $96.2_{\pm2.4}$ | $97.5_{\pm0.6}$ | $49.0_{\pm0.2}$ | $78.1_{\pm1.0}$ |
| INTELLECT-2 (Prime Intellect, 2025) | $75.2_{\pm2.4}$ | $66.3_{\pm6.6}$ | $96.0_{\pm2.7}$ | $97.0_{\pm0.3}$ | $49.8_{\pm0.2}$ | $78.0_{\pm0.8}$ |
| Based on: DeepSeek R1 Distill Qwen 32B (SFT) | | | | | | |
| Distill R1 (DeepSeek-AI, 2025) | $67.0_{\pm1.9}$ | $55.3_{\pm5.7}$ | $96.8_{\pm2.1}$ | $95.1_{\pm0.7}$ | $45.1_{\pm1.7}$ | $73.8_{\pm0.5}$ |
| TinyR1-Preview (Sun et al., 2025) | $73.3_{\pm4.4}$ | $66.0_{\pm6.6}$* | $97.8_{\pm2.2}$ | $97.2_{\pm0.4}$** | $48.5_{\pm0.5}$* | $76.3_{\pm1.2}$* |
| Light-R1 DS (Wen et al., 2025a) | $76.7_{\pm4.7}$* | $68.7_{\pm5.7}$** | $97.0_{\pm1.6}$ | $97.3_{\pm0.1}$** | $48.2_{\pm1.1}$* | $76.9_{\pm0.8}$** |
| Based on: Qwen2.5 32B Instruct (SFT) | | | | | | |
| Qwen (Instruct) (Yang et al., 2024a) | $15.7_{\pm4.7}$ | $13.0_{\pm4.6}$ | $60.8_{\pm5.7}$ | $81.6_{\pm0.5}$ | $40.3_{\pm1.3}$ | $46.7_{\pm1.3}$ |
| Sky-T1-Preview (NovaSky, 2025) | $37.0_{\pm8.1}$** | $22.7_{\pm3.1}$ | $75.8_{\pm3.5}$** | $88.6_{\pm0.5}$** | $39.8_{\pm1.2}$ | $56.3_{\pm0.8}$** |
| Bespoke-Stratos (Bespoke Labs, 2024) | $37.7_{\pm5.9}$** | $26.3_{\pm7.3}$* | $80.7_{\pm3.9}$** | $90.3_{\pm0.3}$** | $43.3_{\pm1.5}$ | $61.5_{\pm1.7}$** |
| LIMO (Ye et al., 2025) | $57.8_{\pm4.7}$** | $46.3_{\pm5.5}$** | $93.0_{\pm2.8}$** | $94.6_{\pm1.2}$** | $45.2_{\pm2.3}$* | $70.5_{\pm0.9}$** |
| s1.1-32B (Muennighoff et al., 2025) | $61.3_{\pm6.9}$** | $49.0_{\pm6.9}$** | $94.2_{\pm3.9}$** | $94.8_{\pm0.5}$** | $46.2_{\pm0.8}$** | $72.4_{\pm0.7}$** |
| OpenThinker (Open Thoughts, 2025) | $68.5_{\pm6.7}$** | $50.3_{\pm6.2}$** | $95.0_{\pm2.9}$** | $95.5_{\pm0.3}$** | $46.8_{\pm1.2}$** | $71.8_{\pm0.5}$** |
| OpenThinker2 (Open Thoughts, 2025) | $71.3_{\pm6.5}$** | $61.7_{\pm5.3}$** | $95.8_{\pm2.9}$** | $96.1_{\pm0.4}$** | $46.6_{\pm0.6}$** | $75.5_{\pm0.6}$** |
| Standalone Models (No tools) | | | | | | |
| GPT-OSS-20B (medium) (OpenAI, 2025b) | $75.7_{\pm5.0}$ | $72.7_{\pm5.4}$ | $97.0_{\pm3.3}$ | $96.5_{\pm0.8}$ | $45.1_{\pm1.1}$ | $77.0_{\pm0.7}$ |

Table 3: **A Standardized and Sober Compilation of Large-Scale (20B-32B) LM-Reasoning Results.** We report Pass@1 accuracy (mean $\pm$ std) of all 32B-scale models across six math reasoning benchmarks under a standardized evaluation setup . RL- and SFT-based variants are evaluated relative to their respective base or instruction-tuned models. Note that * statistically significant at $p < 0.01$; ** statistically significant at $p < 0.001$ (paired t-test relative to base model).

did not carry over significantly to AIME'25 indicating overfitting to existing benchmarks. Only DeepscaleR and FastCuRL demonstrated statistically significant improvements across many benchmarks. Notably, recent models like Nemotron-RR deliver the first recipes with robust improvements over the DeepSeek R1-Distill baseline.

> **Takeaway 1** Most RL-trained variants of the DeepSeek R1-Distill model do not yield meaningful performance improvements (except DeepscaleR and FastCuRL), suggesting that a reliable and scalable RL training recipes were lacking. Nemotron-RR, in contrast, now provides robust improvements.

**RL Training on Qwen2.5 Math and Base Models:** We next analyze RL training applied to the Qwen2.5 Base and Qwen2.5 Math Base models, a trend trying to replicate gains by Deepseek-R1 Zero. Unlike the R1-Distill results, RL training with Oat-Zero, LIMR, and SimpleRL-Zoo consistently produced statistically significant gains over the base model, especially across Math500, Minerva and OlympiadBench benchmarks. This indicates that RL-based approaches can indeed offer substantial improvements given a base model instead of a distilled R1 model. However, these gains are often roughly comparable and sometimes slightly higher than achieved via instruction tuning in the original Qwen papers, suggesting that instruction tuning with additional math data alone may be sufficient to far achieve the current gains from RL methods in this setting. We also observed that the improvements on AIME'24 were also significant, but did not carry over to AIME'25 indicating a troubling overfitting trend.

> **Takeaway 2** While RL-trained methods can often substantially improve base model performance, instruction tuning remains superior (except Open Reasoner Zero), suggesting again that a reliable and scalable RL training recipes are still lacking.

**Effectiveness of Supervised Finetuning.** We assessed supervised finetuning methods like s1.1, Eurus2 Prime, Bespoke Stratos, OpenR1 and OpenThinker models, which further

refine instruction-tuned models using reasoning traces. Supervised methods consistently outperformed the instruct-tuned baseline across all benchmarks (even Minerva) and generalized comparatively well to AIME'25. The performance improvements from OpenThinker series and OpenR1 were especially notable, showcasing the promise of data curation for supervised finetuning. These results underscore the maturity and effectiveness of SFT when training recipes are scaled to large datasets.

> **Takeaway 3** Supervised finetuning on reasoning traces from larger models yields significant, generalizable gains across benchmarks with progress over time successfully replicated — highlighting its robustness and maturity as a training paradigm.

**Scaling to Larger Parameters.** We extend our analysis to 32B scale models in Table 3. Among R1-Distill variants, Light-R1 DS achieves significant performance improvements, demonstrating that supervised finetuning can further improve even strong distilled models. For Qwen2.5-32B base models, SFT approaches like OpenThinker series achieve strong gains (71.3% and 68.5% on AIME'24 respectively) over the instruct model, showcasing the promise of data curation for supervised finetuning. Notably, DAPO - an RL approach, shows improvements over the base model, and showcases strong performance gains even relative to the Qwen Instruct model. The QwQ-32B models demonstrate state-of-the-art performance (76.3% on AIME'24), and INTELLECT-2 shows no meaningful improvement over QwQ-32B despite additional training. These results reinforce our earlier findings: (1) SFT on reasoning traces remains the most reliable approach for performance gains, and (2) scaling to larger models amplifies these benefits, with 32B models achieving substantially higher absolute performance than their smaller counterparts.

> **Takeaway 4** Our findings remain robust to model scale, with SFT methods continuing to dominate RL approaches in both absolute performance and robust gains over baselines.

**Overfitting and Generalization** We now examine the overfitting by comparing performance on AIME'24 versus the more challenging AIME'25. RL-trained models showed a pronounced performance drop between the two, indicating overfitting to the training distribution. In contrast, supervised fine-tuning (SFT) models maintained consistent improvements, suggesting better generalization. Openthinker2 showed significant degradation compared to Openthinker across benchmarks not provided in their blogpost, indicating overfitting via data-curation. This highlights a gap in current evaluation protocols, and a need to assess out-of-distribution generalization for reasoning models.

> **Takeaway 5** Current RL-based approaches are very susceptible to overfitting, emphasizing the need for more rigorous out-of-distribution benchmarks. By comparison, SFT models exhibit stronger generalization and resilience.

## 4    Conclusion

Our study shows that much of the perceived progress in LLM-based reasoning, particularly in mathematical benchmarks, rests on unstable and often non-reproducible foundations. We find that minor differences in sampling parameters, prompt formatting, hardware, and software configurations can lead to major shifts in reported performance—casting doubt on many recent empirical claims. Reinforcement learning methods, while promising in theory, offer at best modest gains in practice and are prone to overfitting, especially on small benchmarks like AIME'24. In contrast, supervised finetuning continues to deliver consistent, generalizable improvements across a wide range of benchmarks and model sizes. To address these challenges, we advocate for standardized, transparent evaluation protocols. Our opensourced framework, complete with Dockerized environments, seed-averaged metrics, and robust answer matching, provides reproducible foundations for future research. We hope this work shifts the focus from leaderboard chasing to methodological rigor—ensuring that future claims of progress in reasoning are both meaningful and measurable.

## Author Contributions

Andreas, Vishaal and Ameya conceived the project. Andreas and Hardik co-led the experiments, with Vishaal and Ameya advising the experimental design. The manuscript was written by Andreas, Hardik, Vishaal and Ameya. Matthias and Samuel provided helpful feedback and advice throughout the project.

## Acknowledgments

The authors would like to thank (in alphabetical order): Matteo Farina, Shyamgopal Karthik, Nikhil Parthasarathy, Shiven Sinha, Joschka Strüber, Thaddäus Wiedemer for helpful feedback on the draft. AH acknowledges funding by the Federal Ministry of Education and Research (BMBF), FKZ: 01IS24079A. HB has received funding from the Digital Europe Programme under grant agreement No 101195233 (OpenEuroLLM). AH, HB and VU thank the International Max Planck Research School for Intelligent Systems (IMPRS-IS) for support. VU also thanks the European Laboratory for Learning and Intelligent Systems (ELLIS) PhD program for support. VU was supported by a Google PhD Fellowship in Machine Intelligence. AP and MB acknowledge financial support by the Federal Ministry of Education and Research (BMBF), FKZ: 011524085B and Open Philanthropy Foundation funded by the Good Ventures Foundation. This work was supported by the Digital Europe Programme under grant agreement No 101195233 (OpenEuroLLM).

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

# Appendix

We now provide thorough details about the benchmark, baselines, and prompts.

## Contents

# A    Recommendations: Which practices to adopt?

We propose a set of best practices informed by our experiments and guided with current research insights:

- **Hardware and Software Stack Standardization:** To promote reproducibility and facilitate future work, we release all code within a Docker container, along with step-by-step instructions for running experiments on Runpod's publicly accessible, on-demand GPU instances. This setup allows any researcher to replicate and extend our results under identical conditions.

- **Variance Estimates:** For small benchmarks (e.g., AIME'24), run evaluations with at least ten random seeds. Report the mean and standard deviation to quantify uncertainty and assess the statistical significance of performance differences.

- **Model-Specific Hyperparameter Optimization:** Tune hyperparameters (such as temperature and `top_p`) separately for each model, then fix them across tasks to ensure consistency and fair comparisons.

- **Context Length and Prompt Template Selection:** Ensure the context length is sufficiently large—especially for models with long reasoning chains—to avoid premature truncation and under-reported accuracy. For instruction-tuned models, always use the appropriate chat template to match the expected input format.

- **Robust Answer Matching:** We strongly recommend using a resilient answer extraction pipeline that handles parsing issues and evaluates expression equivalence, rather than relying on exact string matching. This reduces the likelihood of spurious gains from formatting artifacts.

- **Transparent Evaluation Protocols:** We recommend to release code, prompts, and model outputs, and clearly document the evaluation stack. Report uncertainties (e.g., via standard deviations) and include both quantitative and qualitative analyses to enable thorough and reproducible comparisons.

## B  Do Discovered Phenomena Replicate? A Detailed Analysis.

We further investigate two recently noted phenomena to see if they replicate in our experiments: (1) how response length correlates with performance, and (2) the decline in response diversity following reasoning-focused training.

### B.1  Are Incorrect Responses Longer?

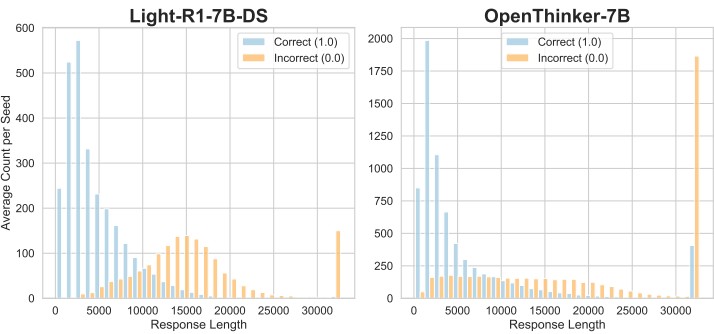

Figure 8: **Response Length vs. Accuracy.** Histogram of correct vs. incorrect responses by response length, averaged over random seeds across AIME24, AIME25, AMC23, MATH500, Minerva and OlympiadBench benchmarks. Longer outputs tend to be more error-prone, even in complete responses not close to the maximum sequence length.

Recent research (Wang et al., 2025) suggests that incorrect answers often have disproportionately long reasoning chains. We first verify whether this finding holds in our setting, and then we explore possible explanations behind the observed variations.

**Do longer responses indicate a higher likelihood of an incorrect answer?** We compare the distribution of response lengths for correct and incorrect answers across 6 datasets (AIME24, AIME25, AMC23, MATH500, Minerva and OlympiadBench) averaged across random seeds for each model. Figure 8 shows histograms of the average number of responses per seed, binned by response length. A clear trend emerges: shorter responses are significantly more likely to be correct, while longer responses become progressively more error-prone. This pattern is consistent across all seeds and is especially pronounced for responses exceeding 10,000 tokens. We now address two questions:

**Q1. Does this pattern hold for both RL- and SFT-trained models?** Yes. We find the trend is consistent across both RL- and SFT-trained models (additional figures provided in Appendix figures 21 and 22 ). We consistently observe that the effect is more pronounced in RL-trained models (displayed on the left) than in SFT-trained models (displayed on the right). As detailed in the Appendix, both the Qwen 2.5 Math base exhibit a slight shift in length, though this shift is notably more evident in R1-distill and subsequent RL-trained models.

**Q2. Is this primarily because of truncated or incomplete responses?** Although responses nearing the 32,000-token limit are almost always incorrect (due to limited context-length), this trend persists even for complete responses which are shorter– Longer responses are associated with a higher likelihood of being incorrect.

> **Takeaway 6** Longer responses correlate with a greater chance of error, response length is a practical heuristic for consensus@k, identifying low-confidence or failed generations.

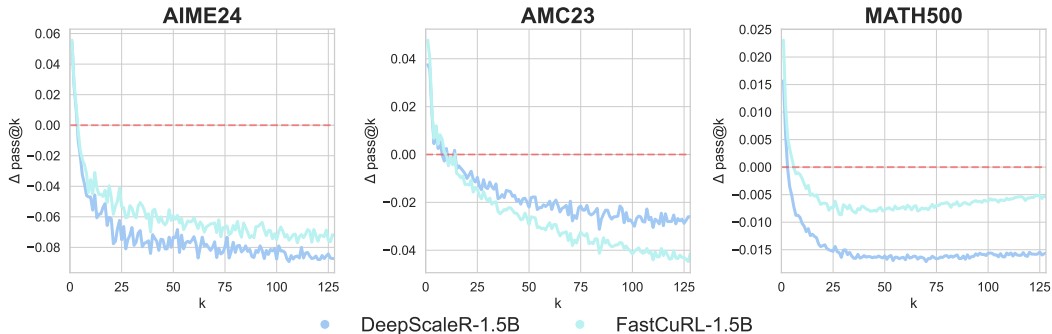

Figure 9: **RL-trained models do show a diversity collapse (Dang et al.).** Across all benchmarks, RL-trained models (DeepScaleR-1.5B and FastCuRL-1.5B) consistently underperform their baseline (DeepSeek-R1-Distill-1.5B) in Pass@k, with all ΔPass@k values being negative. All models were evaluated using the decoding parameters in Table 7.

## B.2 Is There Diversity Collapse in Reasoning Training?

Dang et al. and Yue et al. (2025a) have reported a counterintuitive phenomenon in reasoning models: improvements in Pass@1 achieved through supervised fine-tuning or RL can reduce Pass@k performance due to diminished output diversity—a phenomenon termed *diversity collapse*. Theoretical analyses attribute this collapse to the model concentrating too much probability mass on a single reasoning path, while current decoding strategies fail to recover the lost diversity.

To examine these claims, we compare the Pass@k performance (for $k \in \{1, 2, 3, \ldots, 128\}$) of two RL-trained models (DeepScaleR-1.5B and FastCuRL-1.5B) against their base model DeepSeek-R1-Distill-Qwen-1.5B across all datasets. Figures 9 and 23 show the delta in Pass@k relative to DeepSeek-R1-Distill-Qwen-1.5B.

**Findings.** We do observe a minor diversity collapse. Gains in Pass@1 generally come with regression in Pass@k, though the magnitude of the decay varies.

> **Takeaway 7** Consistent with the diversity collapse hypothesis, the RL-trained models we tested show a regression in Pass@k accuracy.

## C    Extended Related Works

We contextualize our work in broader literature extensively in this section.

**Language Model Reasoning (for Math).** The recent releases of OpenAI-O1 (Jaech et al., 2024) (in September 2024), OpenAI-O3 (OpenAI, 2025a) (in December 2024) and DeepSeek-R1 (DeepSeek-AI, 2025) (in January 2025), have spurred the language modelling community to work on improving the reasoning capabilites of language models. Several popular methods for improving those capabilites have emerged with supervised fine-tuning (SFT) and reinforcement learning (RL) being the two primary methods of interest (Uesato et al., 2022; Lightman et al., 2023; Lyu et al., 2025; Open Thoughts, 2025). Recent works have built upon the DeepSeek-R1 recipe by proposing newer RL algorithms, including LCPO (Aggarwal & Welleck, 2025), REINFORCE++ (Hu, 2025), DAPO (Yu et al., 2025), DPO-VP (Tu et al., 2025), VinePPO (Kazemnejad et al., 2024), CPPO (Lin et al., 2025a), VAPO (Yue et al., 2025b) and GRO (Cai, 2025). To gain a stronger understanding of how to induce mathematical capabilities, other works have conducted significant empirical studies exploring the design space of RL methods (Zeng et al., 2025b; Liu et al., 2025b; Team et al., 2025; Shao et al., 2024), including data scaling trends (Shen et al., 2025), curriculums (Wen et al., 2025a; Roux et al., 2025) and reward design (Gao et al., 2024a; Cui et al., 2025; Ma et al., 2023). Based on the success of these methods, there have also been recent efforts into scaling up reinforcement learning approaches to induce reasoning in domains beyond math, including code (Liu & Zhang, 2025; Xie et al., 2025; Jha et al., 2024; Yu et al., 2024), medicine (Zhang et al., 2025; Sim & Chen, 2024) and other sciences (Su et al., 2025; Yuan et al., 2025; Zeng et al., 2025a). Further, some works also explored scaling up RL-based approaches to modalities beyond just language, including vision (Ma et al., 2025; Meng et al., 2025; Huang et al., 2025; Peng et al., 2025; Chen et al.; Deng et al., 2025; Liu et al., 2025c; Feng et al., 2025; Lin et al., 2025b). In our work, we objectively re-evaluate the claims made by several of these recent works under a standardized lens, and find that many of the reported gains do not hold up strongly when pitted on a level-playing field against well-tuned baselines.

**Sobering Studies on ML Progress.** Machine learning is a field of rapid progress. Due to the lightning speed of papers coming out across the various sub-fields of machine learning, practitioners and researchers often fail to rigorously evaluate algorithmic progress (Hutchinson et al., 2022; Dehghani et al., 2021; Machado et al., 2018; Ghosh et al., 2024; Balduzzi et al., 2018; Liao et al., 2021; Cawley & Talbot, 2010; Lipton & Steinhardt, 2019; Prabhu et al., 2024b; Card et al., 2020; Dror et al., 2018). This has led to several papers showing that simple well-tuned baselines outperform months of progress on a specific sub-field in machine learning, including in continual learning (Prabhu et al., 2024a; 2020), active learning (Cawley, 2011) and test-time adaptation (Press et al., 2023). With the rapid influx of reasoning-based LMs, such statistically rigorous comparisons of models are ever more important—yet, despite the heavy use of RL-algorithms for driving progress in reasoning, there is very little mention of how different methods standardize their evaluations across different factors of variability. RL-algorithms themselves are known to be quite fickle to extremely minor variations including random seeds (Agarwal et al., 2021; Gorsane et al., 2022; Chan et al., 2019; Jordan et al., 2020; Patterson et al., 2024). Some works have even gone as far as suggesting that reliable benchmarking of RL-based methods is computationally infeasible (Jordan et al., 2024). Additionally, other works have demonstrated critical reliability issues in the generalization of frontier models to minor perturbations in the question inputs (Mirzadeh et al., 2024; Nezhurina et al., 2024; Srivastava et al., 2024), the type of tasks tested (Yan et al., 2025b; Petrov et al., 2025; Dominguez-Olmedo et al., 2024; Roberts et al., 2025), metrics used (Liu et al., 2024) and in data-scarce scenarios (Udandarao et al., 2024; Kandpal et al., 2023; Parashar et al., 2024). Given such a volatile landscape, in this work, we aim to level the playing field across recent LM-methods that have been released and provide an objective look on the progress that the reasoning community has made. Our findings, which we discuss in the rest of the paper, are sobering at best.

# D   Variance from Hardware & Software Factors: A Critical Consideration

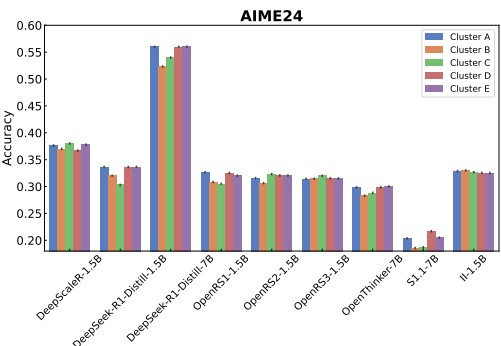
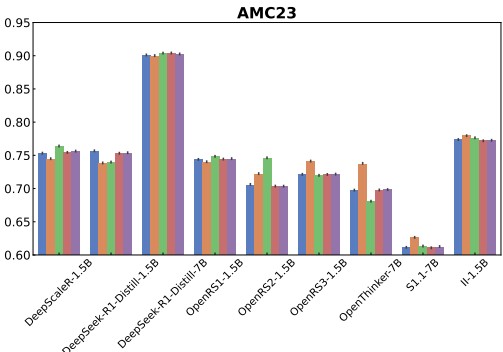

(a) **AIME24.** Significant differences are observed in model performance across compute clusters.

(b) **AMC23.** Similar variability is seen across hardware in AMC23 results.

Figure 10: **Performance variation across compute clusters.** Accuracy differences emerge when the same models are evaluated across compute clusters for both AIME24 and AMC23 datasets—these large differences in performance also persist when evaluating 7B models.

Performance can also vary due to non-obvious factors like hardware and evaluation framework—yet this is rarely acknowledged. Models are often tested on heterogeneous systems and evaluated using different toolchains.For example, S-1.1 (Muennighoff et al., 2025) uses `lm-evaluation-harness` (Gao et al., 2024b), the OpenRS model suite uses `lighteval` (Fourrier et al., 2023), and II-1.5B-Preview uses `evalscope` (Alibaba ModelScope Community) for evaluation.

**Hardware Variation.** We evaluated the same model across five different compute clusters, each with varying GPU types and memory configurations. As shown in Figure 10, performance varied by up to 8% for OpenRS-1.5B and 6% for DeepSeek-R1-Distill-7B on AIME'24, with similar trends observed on AMC'23. While it is known that inference engines such as vLLM can be sensitive to hardware differences (vLLM Contributors, 2024)—and that low-level optimizations in PyTorch or CUDA (PyTorch Contributors, 2024) may introduce non-determinism—our results demonstrate that these effects can measurably impact benchmark accuracy, even when averaging over multiple seeds. In Figure 11, we show that similar discrepancies are observed on MATH500.

**Can we alleviate this?** We conducted extensive experiments to isolate and quantify hardware-induced variability, even when using identical GPU types and software stacks. Our investigation proceeded in three stages, each attempting to eliminate additional sources of variation:

*Stage 1: Standardized Docker across different A100 clusters.* We first evaluated models using our standardized Docker container on A100 GPUs across two platforms: our internal cluster and Runpod (a cloud GPU service). Despite identical hardware specifications and containerized environments, we observed notable performance differences. For instance, Qwen2.5-Math-1.5B achieved 7.3%±3.8 on AIME'24 on our cluster versus 11.3%±3.6 on Runpod; a 4 percentage point gap that exceeds the standard deviation.

*Stage 2: Enforcing CUDA determinism.* Suspecting non-deterministic GPU operations, we enforced strict determinism by setting `torch.use_deterministic_algorithms(True)`, fixing CUDA seeds, and disabling cuDNN benchmarking. Comparing A100 and H100 clusters, variance decreased but persisted: OpenThinker2-7B scored 53.0%±4.6 on A100 versus 57.1%±5.2 on H100 for AIME'24. Even with deterministic algorithms, hardware architectural differences and vLLM's backend variations (vLLM Contributors, 2024) continued to introduce discrepancies.

*Stage 3: Identical hardware, updated stack.* Finally, we updated all components (vLLM, LightEval, math-verify) and ran multiple evaluations on identical hardware. Remarkably, even consecutive runs on the same A100 cluster produced variations: Bespoke-Stratos-7B ranged

from 19.3% to 23.0% on AIME'24 across runs. These persistent differences, despite controlling for hardware, software versions, and random seeds highlight deep-seated sources of non-determinism in modern inference stacks, potentially arising from dynamic kernel selection, memory allocation patterns, or floating-point accumulation order.

As shown in Figure 10, these effects are not merely theoretical but measurably impact benchmark accuracy even when averaging over multiple seeds. This underscores that reproducibility challenges extend beyond simple seed variance to fundamental architectural and systems-level factors.

**Evaluation across different Python frameworks.** Evaluation results can vary based on the framework used, due to differences in prompt templates, inference engines (e.g., vLLM (Kwon et al., 2023)), and response extraction strategies (e.g., MathVerify). For example: `lighteval` is used by OpenRS (Dang & Ngo, 2025), `evalchemy` (Guha et al., 2024) is used by models like OpenThinker and Bespoke-Stratos, other frameworks include `lm-evaluation-harness` (Gao et al., 2024b) and `evalscope` (Alibaba ModelScope Community).

To assess this impact, we compare `lighteval` and `evalchemy`, keeping all other variables fixed: model, dataset, hardware, decoding parameters, and random seeds (3 per model). For a fair comparison, we evaluated two models, DeepSeek-R1-Distill-1.5B and S1.1-7B, at their default `temperature` and `top_p` parameter values on a single GPU. We present results averaged over three seeds for higher robustness. As shown in Table 4, framework-induced differences are generally small (1–2pp) but can still affect model rankings in tightly clustered scenarios.

Overall, our findings underscore that significant performance variations can arise solely from differences in hardware and software configurations, emphasizing the need to standardize for reliable evaluations.

| Model | `lighteval` | `evalchemy` |
|---|---|---|
| R1-Distill-1.5B | 26.6 | 26.6 |
| S1.1-7B | 22.2 | 17.7 |

Table 4: AIME24 across frameworks.

> **Takeaway 6** Hardware and software variations introduce irreducible noise into evaluations, with performance differences of 2-5% persisting even under stringent controls. True reproducibility requires not just seed averaging but also explicit reporting of hardware configurations and acceptance that some variation is inherent to current inference systems.

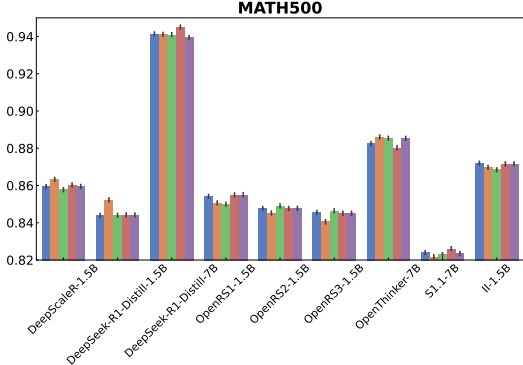

Figure 11: **Performance variation across compute clusters on MATH500.** Differences in GPU type and environment lead to non-trivial shifts in performance, reinforcing the importance of hardware standardization.

# E  Prompt Variants and Template Settings

We provide the exact templates used for our three prompt settings in Table 5: *Math*, *Default*, and *No Template*. These formats are based on the DeepSeek tokenizer but adapted for each model's specific chat template. Our results (in 2.3) indicate that instruction-tuned models are highly sensitive to prompt formatting, with performance degrading significantly when prompts deviate from their training-time structure.

| Prompt | Example |
|---|---|
| Math | `<\|begin_of_sentence\|><\|User\|>Solve the follow-ing math problem efficiently and clearly. The last line of your response should be of the following format: 'Therefore, the final answer is: $\boxed{ANSWER}$. I hope it is correct' (without quotes) where ANSWER is just the final number or expression that solves the problem. Think step by step before answering.\n <\|Assistant\|><think>\n{Question}` |
| Default | `<\|begin_of_sentence\|><\|User\|>{Question} <\|Assistant\|><think>\n` |
| No Template | `{Question}` |

Table 5: **Prompt templates** used in our evaluation. The inclusion or exclusion of structured prompt tokens significantly impacts performance for instruction-tuned models.

# F  Bootstrapping Results on Additional Datasets

To complement our analysis in 2, we present bootstrapped variance results on two additional datasets: AMC'23 and MATH500. As shown in Figures 12 and 13, high variance in Pass@1 persists even when averaging over multiple seeds ($K = 5$), mirroring the trends observed on AIME'24. These results reinforce our conclusion that small benchmark sizes yield unstable estimates and that robust performance reporting requires multiple seed runs. Even for larger Benchmarks, like MATH500 (Figure 14), Minerva (Figure 15) and Olympiad Bench (Figure 16) the estimates remain volative, yet there magnitude is much smaller. An overview of the exact variance values can be found in Table 6.

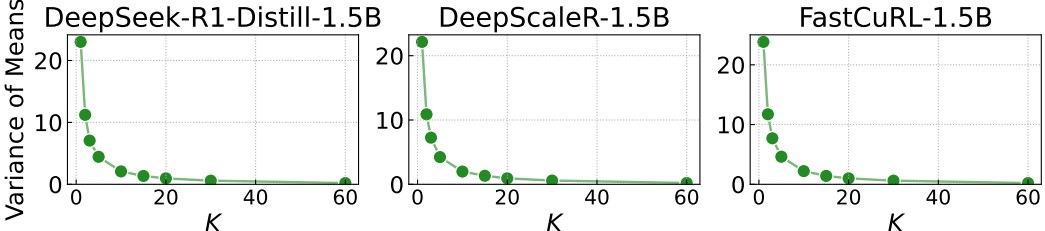

Figure 12: **Variance of mean Pass@1 on AMC'23.** Bootstrapped estimates show substantial variance even with $K = 5$ evaluation runs, highlighting the instability of single-seed evaluations.

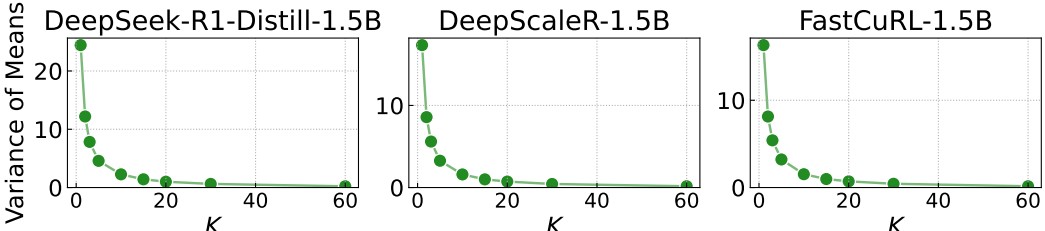

Figure 13: **Variance of mean Pass@1 on AIME'25.** Bootstrapped estimates show substantial variance even with $K = 5$ evaluation runs, highlighting the instability of single-seed evaluations.

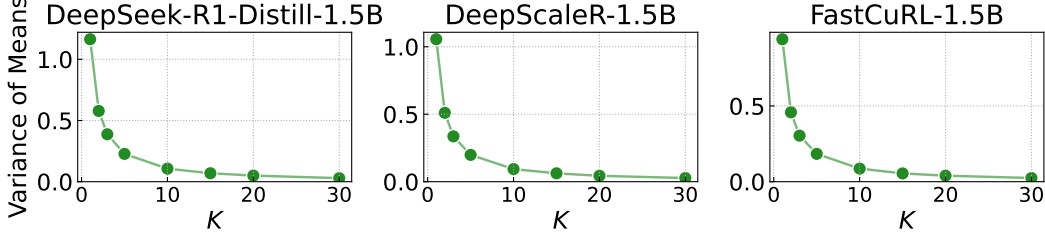

Figure 14: **Variance of mean Pass@1 on MATH500.** Similar to AIME'24 and AMC'23, the estimates remain volatile across seeds. However, since MATH500 is a larger dataset the variance is much smaller than for e.g. AIME'24.

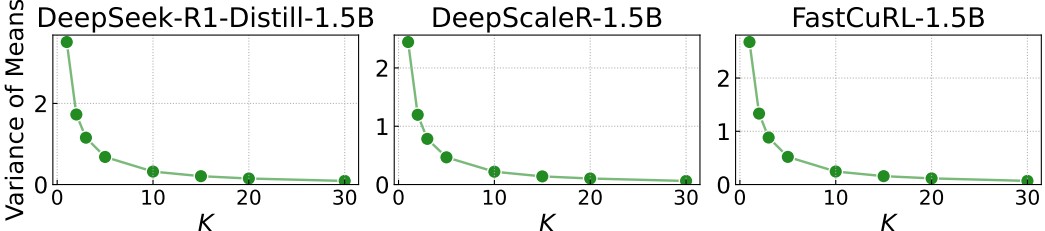

Figure 15: **Variance of mean Pass@1 on Minerva.** Similar to AIME'24 and AMC'23, the estimates remain volatile across seeds. However, since Minerva is a larger dataset the variance is much smaller than for e.g. AIME'24.

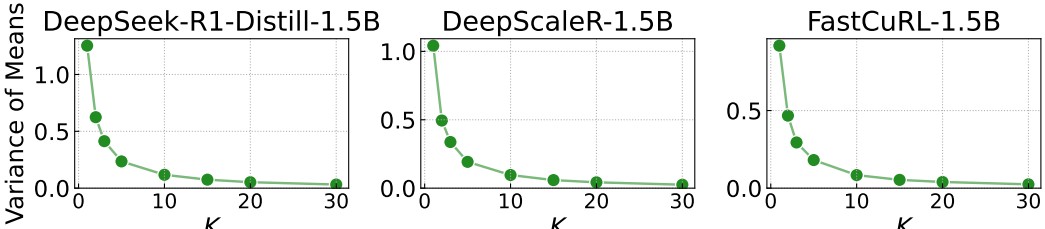

Figure 16: **Variance of mean Pass@1 on Olympiad Bench.** Similar to AIME'24 and AMC'23, the estimates remain volatile across seeds. However, since Olympiad Bench is a larger dataset the variance is much smaller than for e.g. AIME'24.

Table 6: Accuracy variance for different $K$ values across models and datasets

| Model | $K=1$ | $K=2$ | $K=3$ | $K=5$ | $K=10$ | $K=15$ | $K=20$ | $K=30$ | $K=60$ |
|---|---|---|---|---|---|---|---|---|---|
| AIME'24 | | | | | | | | | |
| DeepSeek-R1-Distill-1.5B | 40.044 | 19.841 | 13.269 | 7.762 | 3.754 | 2.409 | 1.686 | 1.054 | 0.365 |
| DeepScaleR-1.5B | 27.452 | 13.477 | 9.117 | 5.342 | 2.495 | 1.587 | 1.149 | 0.705 | 0.246 |
| FastCuRL-1.5B | 28.807 | 14.647 | 9.488 | 5.658 | 2.711 | 1.709 | 1.247 | 0.757 | 0.256 |
| AIME'25 | | | | | | | | | |
| DeepSeek-R1-Distill-1.5B | 24.300 | 11.954 | 7.759 | 4.568 | 2.199 | 1.419 | 1.017 | 0.628 | 0.209 |
| DeepScaleR-1.5B | 17.311 | 8.561 | 5.597 | 3.387 | 1.605 | 0.988 | 0.726 | 0.445 | 0.156 |
| FastCuRL-1.5B | 16.201 | 8.232 | 5.412 | 3.221 | 1.504 | 1.000 | 0.710 | 0.424 | 0.145 |
| AMC'23 | | | | | | | | | |
| DeepSeek-R1-Distill-1.5B | 22.544 | 11.241 | 7.301 | 4.288 | 2.093 | 1.315 | 0.942 | 0.585 | 0.202 |
| DeepScaleR-1.5B | 22.377 | 10.851 | 7.167 | 4.228 | 2.079 | 1.295 | 0.936 | 0.568 | 0.192 |
| FastCuRL-1.5B | 23.671 | 11.841 | 7.649 | 4.605 | 2.149 | 1.404 | 0.980 | 0.594 | 0.211 |
| MATH500 | | | | | | | | | |
| DeepSeek-R1-Distill-1.5B | 1.175 | 0.561 | 0.367 | 0.226 | 0.110 | 0.069 | 0.049 | 0.030 | 0.010 |
| DeepScaleR-1.5B | 1.050 | 0.513 | 0.351 | 0.198 | 0.099 | 0.062 | 0.044 | 0.027 | 0.009 |
| FastCuRL-1.5B | 0.939 | 0.447 | 0.305 | 0.179 | 0.085 | 0.055 | 0.040 | 0.024 | 0.008 |
| Minverva | | | | | | | | | |
| DeepSeek-R1-Distill-1.5B | 3.485 | 1.741 | 1.143 | 0.665 | 0.326 | 0.206 | 0.149 | 0.091 | 0.031 |
| DeepScaleR-1.5B | 2.385 | 1.179 | 0.769 | 0.452 | 0.220 | 0.143 | 0.102 | 0.061 | 0.022 |
| FastCuRL-1.5B | 2.703 | 1.347 | 0.876 | 0.527 | 0.244 | 0.161 | 0.113 | 0.069 | 0.024 |
| Olympiad Bench | | | | | | | | | |
| DeepSeek-R1-Distill-1.5B | 1.269 | 0.622 | 0.427 | 0.241 | 0.118 | 0.076 | 0.053 | 0.032 | 0.011 |
| DeepScaleR-1.5B | 1.018 | 0.505 | 0.334 | 0.195 | 0.093 | 0.059 | 0.043 | 0.026 | 0.009 |
| FastCuRL-1.5B | 0.908 | 0.463 | 0.305 | 0.179 | 0.086 | 0.054 | 0.039 | 0.024 | 0.008 |

## G  Temperature and top-p: Additional Analysis

We additionally investigate the impact of the temperature and `top_p` hyperparameter as prior works often employ different temperature and `top_p` settings when comparing the same model. To isolate the impact of varying temperature and `top_p`, we averaged pass@1 across seeds and compute variation of this estimate across temperature and `top_p` in a boxplot. Figure 17 and 18 show the performance variation. We see that temperature-induced and `top_p`-induced fluctuations not only affect performance estimates but also introduce substantial variability in performance itself, which can lead to unfair comparisons when evaluating the same model across different temperatures.

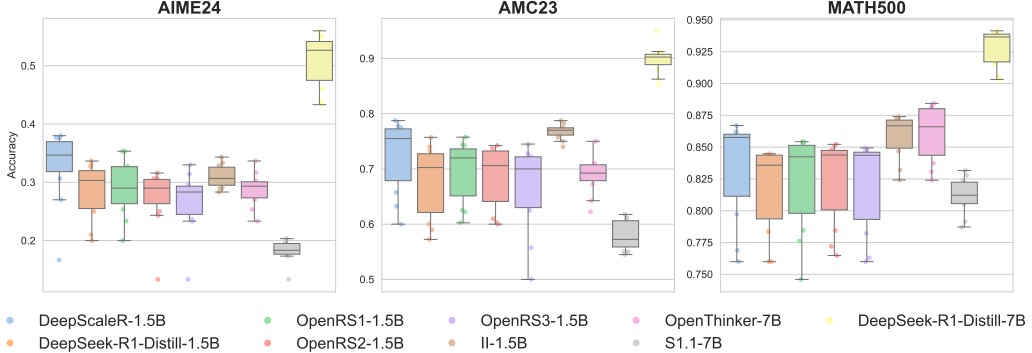

Figure 17: **Accuracies vary significantly across temperature values.** Across nine different models and three datasets, we observe consistently large variations in performance (upto 15%) induced by changing the temperature. Results were obtained by varying the temperature from 0 to 1 in increments of 0.1, while holding `top_p` constant at 0.9.

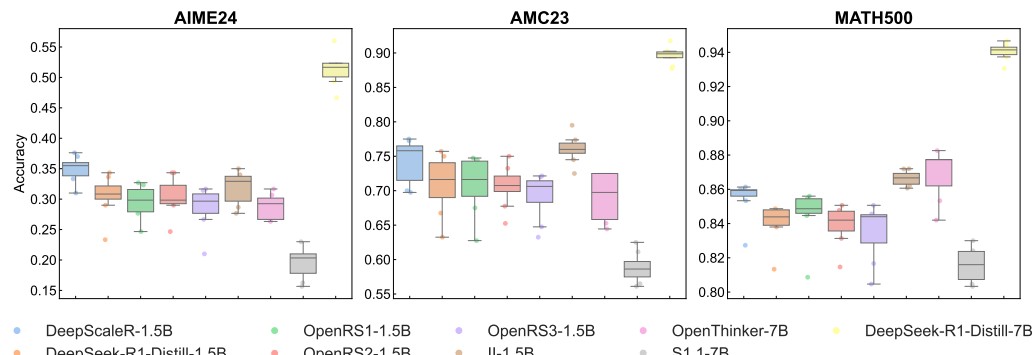

Figure 18: **Accuracies vary significantly across `top_p` values.** Across nine different models and three datasets, we observe consistently large variations in performance (upto 8%) induced by changing the `top_p` value. Results were obtained by varying `top_p` from 0 to 1 in increments of 0.1, while holding the temperature constant at 0.8.

# H    Effect of Output Length Limits

We further explore how varying `max_new_tokens` impacts model accuracy. Figures below compare OpenRS-series models (with 131,072-token context windows) and Open-Thinker/S1.1 models (with 32,768-token limits).

Figure 19 shows that OpenRS models are highly sensitive to this parameter—shortening outputs results in clear accuracy drops. Similarly, Figure 20 reveals the same pattern for OpenThinker-7B and S1.1-7B, despite their smaller context lengths. In both cases, premature truncation leads to incomplete reasoning chains and incorrect answers, confirming the importance of setting appropriate generation limits.

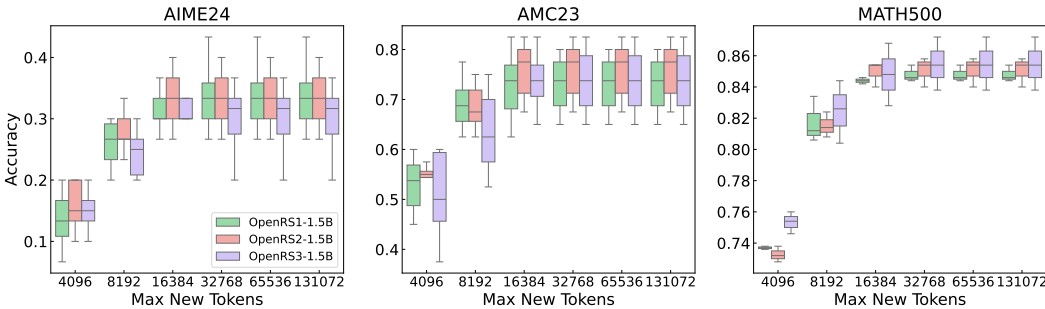

Figure 19: **Impact of `max_new_tokens` on OpenRS models.** Models with long context support (131,072 tokens) experience degraded performance when `max_new_tokens` is set too low.

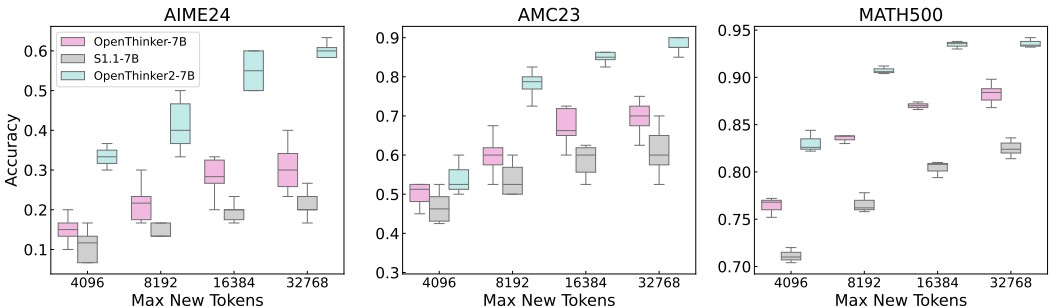

Figure 20: **Impact of `max_new_tokens` on OpenThinker and S1.1 models.** Despite shorter context limits (32,768 tokens), performance still degrades noticeably when output length is constrained.

# I   Response Length vs. Accuracy — Per-Model Breakdown

To supplement the aggregated results shown in Figure 8, we include detailed histograms for each individual model in the appendix. These plots show the distribution of correct and incorrect responses across response lengths, averaged over random seeds. Due to the number of models analyzed, we split the results into two figures for clarity.

Figures 21 and 22 reveal that the overall trend observed in the main paper holds consistently across nearly all models: incorrect responses tend to be longer than correct ones.

These results reinforce the idea that excessively long outputs often indicate failure modes such as hallucinated reasoning, verbose overthinking, or degenerate loops. Importantly, this correlation persists well below the maximum sequence length, ruling out truncation as the sole cause. Across all models, longer responses are a consistent marker of incorrect outputs, making response length a useful signal for detecting low-confidence or erroneous reasoning chains.

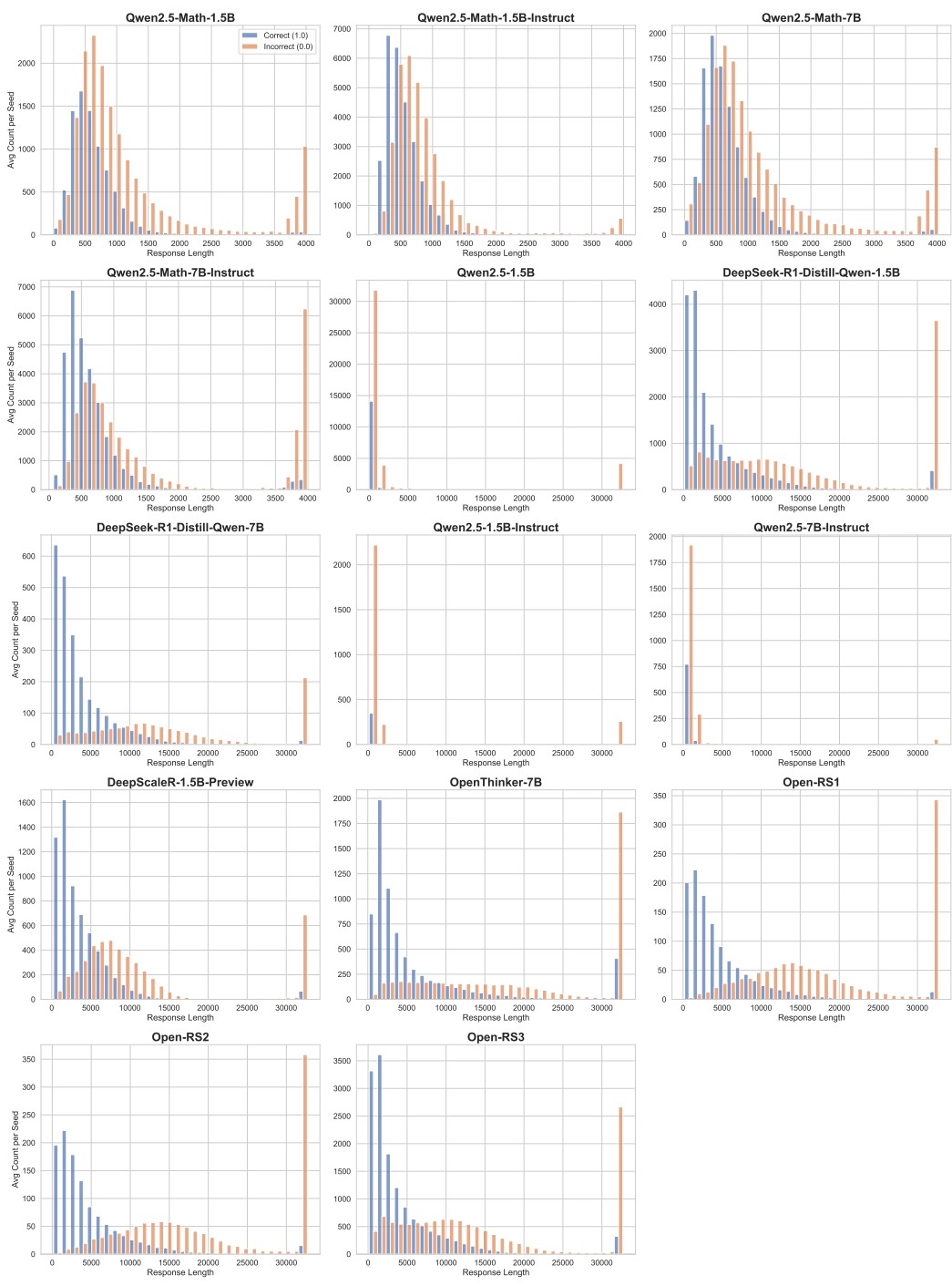

Figure 21: **Response Length vs. Correctness — Models (1/2).** Average number of correct and incorrect responses across response length bins for a subset of models. Longer responses consistently correlate with incorrect predictions.

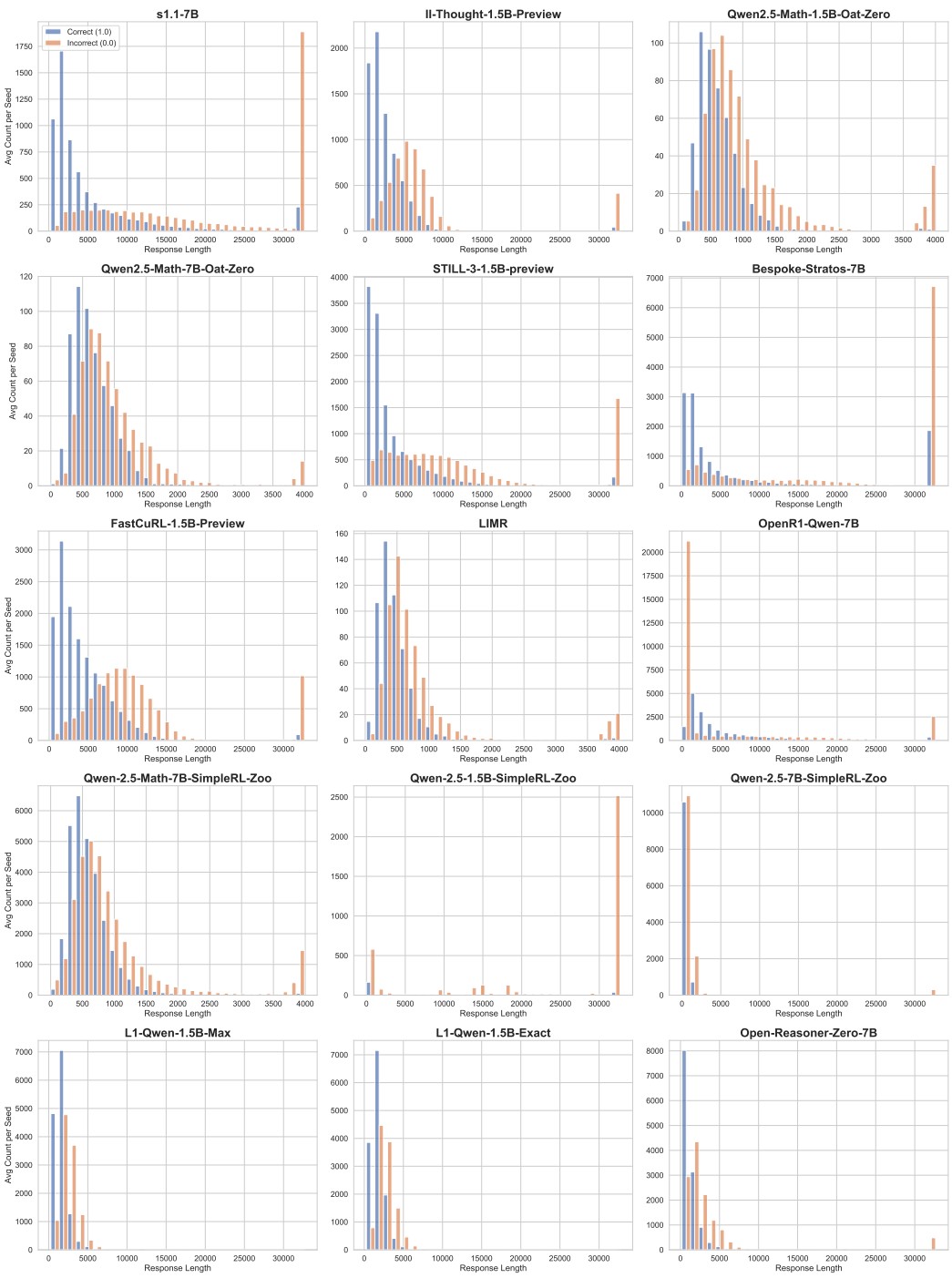

Figure 22: **Response Length vs. Correctness — Models (2/2).** Continuation of model-wise response length analysis. The same trend holds across the remaining models, with incorrect answers being disproportionately long.

## J Diversity Collapse

In addition to the diversity collapse discussed in section B.2 and Figure 9 the results for additional datasets are shown in Figure 23. For DeepSeek-R1-Distill-1.5B (SFT-trained) we could not replicate a diversity collapse as shown in Figure 24.

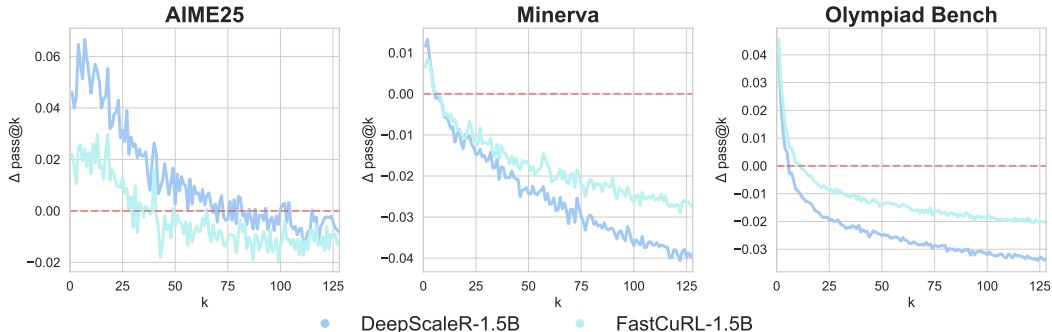

Figure 23: **RL-trained models do show a diversity collapse (Dang et al.).** We report the delta between Pass@k of RL-trained models (DeepScaleR-1.5B and FastCuRL-1.5B) and their corresponding baseline (DeepSeek-R1-Distill-1.5B). We observe a diversity collapse (Δpass@k is below zero). All models were evaluated using the decoding parameters listed in Table 7.

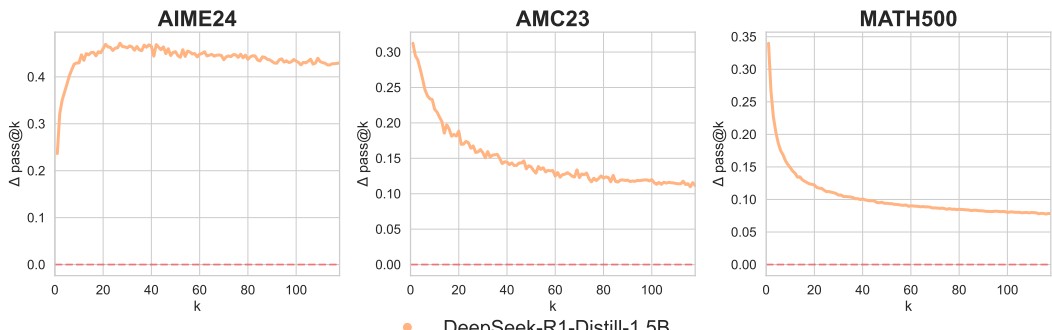

Figure 24: **SFT-trained models do not exhibit diversity collapse.** Across benchmarks, DeepSeek-R1-Distill-1.5B (SFT-trained) exceeds its baseline, Qwen2.5-Math-1.5B, in ΔPass@k. All evaluations used the decoding parameters in Table 7.

## K   Optimal Decoding Parameters

Empirically, the decoding parameters in Table 7 consistently produced optimal performance for the models we evaluated.

Table 7: Optimal temperature and top-p settings for various models

| Model Name | Temperature | Top-p |
|---|---|---|
| agentica-org/DeepScaleR-1.5B-Preview | 1.0 | 0.7 |
| bespokelabs/Bespoke-Stratos-7B | 1.0 | 0.9 |
| deepseek-ai/DeepSeek-R1-Distill-Qwen-1.5B | 0.9 | 0.7 |
| deepseek-ai/DeepSeek-R1-Distill-Qwen-7B | 0.9 | 0.7 |
| hkust-nlp/Qwen-2.5-7B-SimpleRL-Zoo | 0.5 | 0.5 |
| Intelligent-Internet/II-Thought-1.5B-Preview | 1.0 | 0.5 |
| knoveleng/Open-RS1 | 1.0 | 0.6 |
| knoveleng/Open-RS2 | 0.9 | 0.8 |
| knoveleng/Open-RS3 | 0.5 | 0.9 |
| l3lab/L1-Qwen-1.5B-Exact | 0.5 | 0.7 |
| l3lab/L1-Qwen-1.5B-Max | 0.7 | 0.9 |
| Nickyang/FastCuRL-1.5B-Preview | 1.0 | 0.7 |
| Open-Reasoner-Zero/Open-Reasoner-Zero-7B | 0.7 | 0.5 |
| open-thoughts/OpenThinker-7B | 0.8 | 0.95 |
| qihoo360/Light-R1-7B-DS | 0.7 | 1.0 |
| RUC-AIBOX/STILL-3-1.5B-preview | 1.0 | 0.6 |
| simplescaling/s1.1-7B | 1.0 | 0.9 |
| GAIR/LIMR | 0.6 | 0.6 |
| open-r1/OpenR1-Qwen-7B | 0.8 | 0.9 |
| Qwen/Qwen2.5-1.5B-Instruct | 0.2 | 1.0 |
| Qwen/Qwen2.5-7B-Instruct | 0.4 | 0.95 |
| Qwen/Qwen2.5-Math-1.5B | 0.7 | 0.5 |
| Qwen/Qwen2.5-Math-7B | 0.5 | 0.5 |
| sail/Qwen2.5-Math-1.5B-Oat-Zero | 0.6 | 0.6 |
| sail/Qwen2.5-Math-7B-Oat-Zero | 0.6 | 0.6 |

