# OpenReview forum: "A Sober Look at Progress in Language Model Reasoning: Pitfalls and Paths to Reproducibility"
_colmweb.org/COLM/2025/Conference — COLM 2025_

### Official Review · Reviewer_xRby · 2025-05-12

**Rating:** 7
**Confidence:** 4
**Ethics Flag:** 1

**Summary:**

The paper assesses robustness in performance of both RL and SFT models in reasoning benchmarks, with a focus on maths. They investigate in particular the dependence on seeds, prompts, hardware, and sampling parameters.

They show that, also due to the small size of many benchmarks, results are brittle. In particular, several RL frameworks do not show consistent improvements over baseline models and can also overfit.

The authors provide a standardised framework, including optimized parameters per model, that allows fair and consistent, reproducible evaluation, including variance and statistical significance analysis.

**Questions To Authors:**

See above.

**Reasons To Accept:**

--- The paper is very systematic and its results are well-supported.
--- Although some of the results (such as variation in small benchmarks, or seed dependence) are known in ML and statistics in general, it is valuable to be reminded of them in the LLM reasoning context.
--- The paper investigates several aspects such as decoding parameters and hardware configurations that are less often reported
--- The evaluation framework, if published, would greatly promote reproducibility and help to alleviate overclaims in other papers.

**Reasons To Reject:**

--- Some of the results such as variation in small benchmarks are more or less to be expected. Here I also miss citation of papers such as Card et al (EMNLP 2020, with little power comes great responsibility) or Rotem Dror, Gili Baumer, Segev Shlomov, and Roi Reichart. 2018. The hitchhiker’s guide to testing statistical significance in natural language processing. In Proceedings of ACL. It seems that the NLP community needs to rediscover basic statistical facts on evaluation every 5 years or so.

--- In line 2019 and following lines you say that for example FastCurl yields only modest improvements over the baseline on AMC 23, Math500 and OlympiadBench. I cannot see this from Table 3 where there are very clear significant gains of the model especially for Math500 and OlympiadBench with no overlapping confidence intervals.  I think here you need to be more nuanced. I am therefore not sure that one of your claims of the lack of progress via RL is completely true.

--- Why are there more models in Table 3 than in Table 1? It would be good to see the information in table 1 for all models.

---

> ### Author Response · Authors · 2025-05-31
> **Rebuttal**
>
> We thank the reviewer for their thoughtful comments and positive evaluation of our work. We appreciate the recognition of the value of our standardized framework and are grateful for the constructive suggestions. Below, we address the specific concerns raised:
>
> **Missing Citations**: In our revised manuscript, we added a dedicated Related Work section -- adding citations from NLP literature, including the ones suggested by the reviewer alongside from RL literature -- both of which were relevant to this work for thorough coverage.
> - We provide a reference image: https://imgur.com/a/l4BcKEy
>
> **FastCurl**: Thanks for catching this! We agree that FastCuRL similarly shows significant gains on MATH500 and Olympiadbench benchmarks. We should formulate this more nuanced and have revised this in the manuscript as follows:
>
> > Only DeepscaleR and FastCuRL demonstrate robust, significant improvements across MATH500 and Olympiadbench benchmarks.
>
> However, they do not show significant improvements across AIME (AIME'24 for DeepScaleR and AIME'25 for FastCuRL), AMC or Minerva unlike the SFT variants in contrast. In this context, we say that a reliable and scalable RL training recipe is still lacking.
>
> **Missing Models in Table 1:** We agree that aligning the models across Table 1 and Table 3 would improve clarity. We will update Table 1 to include metadata for all models where this information could be reliably recovered. However, in some cases, the original papers do not provide sufficient detail about their evaluation configurations (e.g., decoding parameters or frameworks), which limits our ability to populate Table 1 uniformly. In our best practices, we recommend that future work publish full evaluation configurations to improve reproducibility and comparability.

---

> > ### Comment · Reviewer_xRby · 2025-06-04
> >
> > I am happy with the answers. I think it Table 3 would be edited a bit better to also alleviate Reviewer 1's concerns, this paper is indeed quite strong and a definite acceptance.

---

### Official Review · Reviewer_ja19 · 2025-05-12

**Rating:** 8
**Confidence:** 4
**Ethics Flag:** 1

**Summary:**

The authors measure performances of multiple LMs ranging from 1.5B ~ 7B parameter sizes and provide practical guidelines for reproducible evaluation. Specifically, the authors test the effect of (1) running the evaluation multiple times (bootstrapping), (2) setting different temperature and top p hyper-parameters, (3) using different hardware, (4) different prompt formats, and (5) different context lengths (max new tokens hyper-parameter).

**Questions To Authors:**

1. Figure 1,2,3,4 gives important guidelines for practitioners to take into account when running evaluations, yet, is there a reason why the authors haven't checked whether the results are statistically significant? In addition to reporting that the variance tends to decrease (Figure 2) and whisker plots (Figure 1,3,4), maybe you could also run a paired t-test or report confidence intervals for ablating different configurations.

2. While reporting the LM's performance with a single run is a bad practice, I think that the reason why it is hard to reproduce results in AIME24, AMC23, and MATH-500 also stems from the fact that the instances consisting the benchmark are too small. While it might be out of scope, could the authors also provide guidelines for benchmark creators as well?

**Reasons To Accept:**

While some might view that the findings are too trivial, I think that the authors pose an important message that are supported with practical guidelines that make this paper suitable to be presented at COLM.

**Reasons To Reject:**

There are no reasons to reject this paper, yet, it would be great if the authors could answer the questions I have on the "Questions to Authors" tab.

---

> ### Author Response · Authors · 2025-06-03
> **Rebuttal**
>
> We sincerely thank the reviewer for their time and positive assessment of our work. We appreciate their recognition of the importance of our findings and their practical value for the community. Below, we address the reviewer’s questions in detail:
>
> 1. **Statistical Significance Testing:** We agree that formal significance testing is a valuable addition. We have updated Table 3 and indicate statistical significance relative to the base model for $p<0.01$ and $p<0.001$ with $^{*}$ and $^{**}$ respectively using a one-sided paired t-test.
>    We provide a reference image for showcasing the update: https://imgur.com/a/EIPqfHb
>
> 2. **Guidelines for Benchmark Construction:** We fully agree that small benchmark sizes contribute significantly to the instability of results. While our variance estimates suggest that running evaluations with at least ten random seeds can mitigate this instability, the root issue remains the lack of statistical power due to the small number of test samples. We believe that establishing minimum design criteria -- such as larger sample sizes and more diverse problem distributions -- would benefit the field. That said, we also acknowledge that many widely used benchmarks in reasoning, such as AIME or AMC, are derived from human exams and competitions, which naturally impose size constraints. As models become increasingly capable and data contamination concerns require stricter temporal filtering, it may be necessary to embrace and adapt evaluation protocols to work effectively with these small-scale but high-value benchmarks.

---

> > ### Comment · Reviewer_ja19 · 2025-06-09
> >
> > Thank you for your response!
> >
> > As mentioned in my reviews, I think the paper conveys an important message for the research community and hope that this paper is presented at the main conference. I will keep my positive score as it is!

---

### Official Review · Reviewer_2SGm · 2025-05-13

**Rating:** 6
**Confidence:** 4
**Ethics Flag:** 1

**Summary:**

This paper studies the volatility of evaluation practices in reasoning research. Through a comprehensive investigation of open models, the authors diagnose significant variance in evaluation across different configurations of random seed, temperature, top-p, hardware specifications, and prompt format. They then propose a standard procedure for evaluation standardization, which leads to findings about the realistic effectiveness of RL and SFT algorithms.

**Questions To Authors:**

I'm not sure what Figure x on line 124 refers to.

**Reasons To Accept:**

The call for a more rigorous evaluation is important and timely. I do believe that the first part (section 2) of the investigation which highlights the numerous parameters that can affect evaluation is comprehensive.

**Reasons To Reject:**

Section 3.3, which comprises a very important part of the paper narrative, is not easy to follow and understand. Table 3 is overwhelming with no markings to help digest it. It's hard to understand what numbers the text is trying to refer to while reading. The table's caption says, "Best scores per benchmark are highlighted." But that's not the case. The text also says, "Open Reasoner Zero 7B was the only RL-trained model to consistently outperform the instruct-tuned baselines across all benchmarks." But doesn't it underperform Qwen (Instruct) on Minerva? In general, RL and SFT systems need to be easier to tell apart in the table.

As another example of something that throws off the reader, line 232 says, "Once again, improvements seen on AIME'24 did not generalize to AIME'25," which is completely out of place. It seems to belong to the next paragraph under *Overfitting and Generalization*. As it's hard for me to connect the table and the text, I also find it difficult to confirm if it corroborates the authors' claims.

Also, a dedicated "Related Work" section is missing. While the authors refer to related work throughout the text, given that similar sensitivities in evaluation have been studied elsewhere, a consolidated "Related Work" section would be helpful.

---

> ### Author Response · Authors · 2025-05-31
> **Rebuttal**
>
> We thank the reviewer for their detailed and constructive feedback. We appreciate the reviewer’s positive assessment of Section 2 and the importance of rigorous evaluation. We agree with the concerns raised and have revised the manuscript accordingly.
>
> **Related Work**: We have added a dedicated “Related Work” section for more thorough coverage.
> - We provide a reference image: https://imgur.com/a/l4BcKEy
>
> **Table 3 Formatting**: We have separated Table 3 into sections for RL-based/SFT-based models as per request (alongside expanding number of seeds, and using default chat template). This does improve readability, thanks -- providing the clear interpretation: Accuracies of methods can be compared to the base in each group, and the reference improvements traced to RL/SFT algorithms. We removed the highlight as identifying SOTA is not the intention here and might distract.
>   - We provide a reference image for showcasing the update: https://imgur.com/a/EIPqfHb
>
> We hope referencing this new table will make it far easier to corroborate with the text and judge claims. Thanks for this suggestion!
>
> **Other fixes in errors/typos & readability**
>
> - *Section 3.3 Readability*: We made minor edits to Section 3.3 to improve its narrative structure, providing better clarity.
> - *Line 232*: We removed it to avoid confusion. The next paragraph under *Overfitting and Generalization* already repeats the point.
> - *Open Reasoner Zero*: Thanks, we corrected it to "Open Reasoner-Zero-7B was the only RL-trained model to outperform the instruct-tuned baseline by significant margins across **most** benchmarks."
> - *Broken Reference*: Thanks! Line 124 refers to Figure 3. Fixed!
>
> We hope we have addressed the major concerns of the reviewer, and are happy to answer any further questions/concerns. We look forward to a fruitful reviewer-author discussion phase.

---

> > ### Author Response · Authors · 2025-06-08
> > **Rebuttal Summary and Request for Final Feedback**
> >
> > As the discussion period is ending, we kindly ask the reviewer to let us know whether our response sufficiently addresses their concerns -- we have added new experiments, introduced a related work section, redesigned Table 3, and updated the draft to fix all reported clarity issues. We welcome any additional feedback on our rebuttal. We thank the reviewer again for their time and thoughtful review.

---

> > > ### Comment · Reviewer_2SGm · 2025-06-09
> > >
> > > Thanks for sharing the updates you've made. I appreciate the effort. Can you clarify what else changed besides using 10 seeds that resulted in drastic changes in some numbers? I've included a before and after example here: https://imgur.com/a/uscf5qS. The base, in particular.
> > >
> > > I agree with other reviewers that the message of the paper is important. I'm just trying to fully understand if and how the numbers back that message up.

---

> > > > ### Author Response · Authors · 2025-06-10
> > > >
> > > > Thank you for the comment. Since the original release, we've improved the answer extraction pipeline. We now use a two-stage extraction method: We apply the standard LightEval LaTeX-based extraction (prioritizing \boxed{} matches) and in addition we use LightEval's expression-based matching (reference code from LightEval [here](https://github.com/huggingface/lighteval/blob/main/src/lighteval/metrics/utils/extractive_match_utils.py#L57)) This seems to particularly improve performance for most models. However, this results in quite drastic changes in performance in base models, which we hypothesise are much more unstable due to not following formatting instructions reliably. A similar phenomenon is documented by [Liu et al., 2025](https://arxiv.org/abs/2503.20783) ("Understanding R1-Zero-Like Training: A Critical Perspective")

---

> ### Comment · Reviewer_2SGm · 2025-06-10
>
> Thank you for getting back to me, especially on the day of the deadline. I’m fine being overruled by the other reviewers. I’ve updated my score (from 4 to 6, which is the highest I’m willing to go) to acknowledge your responses and efforts during the response period.
>
> But I’ll also say this for the meta-review period: these are all the kinds of details that should be explicitly mentioned in the paper so that the reader can assess if the systems were treated equally, among other things, given how sensitive they can be. Otherwise, it’ll strongly undermine your message.

---

> > ### Author Response · Authors · 2025-06-11
> >
> > Thank you for the feedback -- we will make sure to add these details to the appendix of the updated draft. We commit to release all sample-wise results, including metadata such as decoding parameters, along with the code used for answer extraction and verification. This will allow the community to inspect the prompts, model responses, and extraction strategies in detail. To ensure reproducibility and ease of use, we also plan to package the entire evaluation setup as a Docker image, so that anyone can run standardized evaluations on new models with minimal setup.
> > Thanks again for your time, we greatly appreciate your engagement.

---

### Author Response · Authors · 2025-05-31
**Summary of Changes**

We thank all reviewers for their time and thoughtful feedback. In response to multiple overlapping requests, we have prepared several additions and clarifications that we include in this general comment for convenience. These changes reflect our efforts since the review deadline to improve the clarity, completeness, and reproducibility of the paper.

Specifically, we include the following attachments:

**Dedicated Related Work Section** — now consolidated and expanded to properly contextualize our contributions.
- Reference image: https://imgur.com/a/l4BcKEy

**Redesigned Table 3** — Now split into separate sections for RL and SFT models, with clearer visual distinctions.
Changes:
- We increased number of seeds to 10 for AIME'24, AIME'25 and AMC'23 for better mean estimation, and standardized evals to use the default chat template (Table 4 of Appendix) across all evaluations.
- We added new models, and are committed to expanding our evaluation to include 32B models in the coming days.
- We indicate statistical significance relative to the base model for $p<0.01$ and $p<0.001$ with $^{*}$ and $^{**}$ respectively using a one-sided paired t-test
- Reference image: https://imgur.com/a/EIPqfHb

---

### Decision · Program_Chairs · 2025-07-08

**Decision:**

Accept

**Comment:**

This work investigates the stability of results of reasoning benchmarks, with a focus on math benchmarks. The authors investigate the robustness of results with respect to factors like random seeds, prompts, hardware, and decoding parameters. They then propose a standardized framework for how to perform more reliable evaluation of reasoning abilities.

All reviewers agreed that this was a valuable and very well executed study highlighting many of the shortcomings in evaluations of reasoning abilities.

While a lot of these findings in general (beyond the assessment of reasoning benchmarks) are not extremely new, they nicely highlight that the field should take better care in performing evaluations and comparisons between models. Furthermore, all reviewers found the proposed framework useful.

The reviewers made several valuable suggestions regarding presentation, especially regarding Section 3, which should be incorporated in the final version of this paper.